# GRIDAGENT: A 2D GRID-BASED GAME BENCHMARK FOR MULTIMODAL LARGE LANGUAGE MODELS

## ABSTRACT

Multimodal Large Language Models (MLLMs) integrate the linguistic capabilities of LLMs with the ability to process multimodal data, enabling them to address a wider array of tasks. However, a comprehensive and standardized benchmark for evaluating MLLMs' complex visual reasoning performance in multimodal tasks has yet to be established. We introduce GridAgent, a versatile 2D grid-based framework that serves as a benchmark for assessing five essential capabilities of MLLMs: **execution, perception reasoning, memory, learning, and planning**. The framework includes twelve unique game tasks specifically designed to avoid overlap with the model's pre-training corpus. Each task targets at least one core competency and is enriched with diverse semantic information. Additionally, the game layouts are randomly generated, ensuring a more rigorous and authentic assessment of the MLLMs' capabilities. Experimental results indicate that although certain MLLMs excel in specific capabilities, none exhibit a comprehensive skill set comparable to the human baseline. Our work can be seen at: `https://iclr2025gridagent.github.io/GridAgent-website/`.

## 1 INTRODUCTION

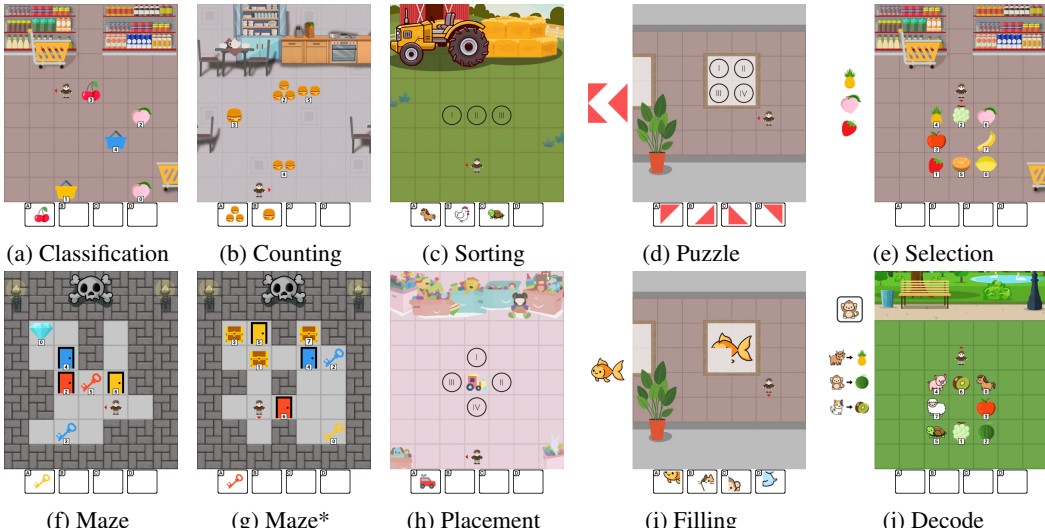

Figure 1: The various game preview in GridAgent, arranged from top left to bottom right are: Classification, Counting, Sorting, Puzzle, Selection, Maze, Maze*, Placement, Filling and Decode.

Large language models (LLMs) have demonstrated significant success across language-based tasks (Brown et al., 2020; Sharan et al., 2023), laying a strong foundation for advancements in artificial intelligence. Following this success, multimodal large language models (MLLMs), which integrate multiple data modalities such as images (Wang et al., 2024e), videos (Cai et al., 2024; Wang

et al., 2024b) and speech (Çoban et al., 2024; Cappellazzo et al., 2024), are also experiencing rapid growth and development. By fusing diverse information sources, MLLMs enable AI to learn and reason (Gao et al., 2024) across different modalities, bringing it closer to human-like cognition (Du et al., 2024). This cross-modal learning capability (Li et al., 2022) enhances the flexibility and generalization of AI systems, potentially leading to more comprehensive decision-making (Chen et al., 2024a) and transfer learning (Wu et al., 2024b), which are viewed as important steps toward the realization of artificial general intelligence (AGI) (Morris et al., 2024).

While research on large LLMs has increasingly concentrated on evaluating and improving their advanced capabilities, such as reasoning (Hendrycks et al., 2020; Shinn et al., 2023), problem-solving (Zhong et al., 2023; Park et al., 2023), and long-tern planning (Hong et al., 2023; Ahn et al.), much of the MLLM research still primarily focuses on the effectiveness of these models in understanding images and other non-text modalities. There is currently no comprehensive and unified benchmark to evaluate whether MLLMs, after acquiring such multimodal understanding, can demonstrate the same remarkable capabilities across multimodal tasks as LLMs do in natural language processing (Radford et al., 2019; Nasution & Onan, 2024). The absence of such a benchmark hampers the ability to systematically assess and compare the performance of different MLLMs, making it challenging to identify their strengths and weaknesses. This lack of standardization may also slow down progress in the field, as researchers and practitioners lack clear metrics to guide their work, ultimately hindering the development of more capable and versatile AI systems.

In this work, we introduce GridAgent, a benchmark specifically designed for evaluating MLLM agents. To assess the gap between MLLMs and human-level capabilities (Lake et al., 2016; Xie et al., 2024; Koh et al., 2024), we draw inspiration from the widely recognized human intelligence test (Smith & Gasser, 2005), Wechsler Intelligence Scales (Guertin et al., 1966; Zhu et al., 2004; Park & Demakis, 2017), and propose five essential abilities that MLLMs require: **execution, perception reasoning, memory, learning and planning**. We design a series of tasks to evaluate the individual capabilities of MLLMs, alongside composite tasks that assess more complex abilities, such as the integration of memory and planning (Xu et al., 2024b). Our tasks also feature varying difficulty levels to test the performance of more advanced MLLMs. Additionally, we pre-configure a variety of background images containing semantic information and incorporate different types of game items to construct specific semantic scenes. To ensure variability, each game is initialized with a randomized layout. This multi-dimensional evaluation framework aims to assess the generalization and robustness of MLLMs, offering a more comprehensive measure of their performance beyond reliance on pre-training data (Zhang et al., 2024b).

Experimental results show that GPT-4o (OpenAI, 2024) outperforms other models, achieving the highest success rate across most tasks and demonstrating a significant advantage in planning-related tasks. Other models exhibit strengths in specific areas; for instance, Qwen2 (Yang et al., 2024) excels in abstract perception reasoning tasks. However, no MLLM currently possesses a comprehensive capability comparable to that of humans, with some models even performing worse than random baselines, falling well below human benchmarks.

Our contributions are as follows:

- We propose five key abilities, inspired by the Wechsler Intelligence Test, to evaluate both the individual and composite capabilities of MLLMs.
- We introduce GridAgent, the first unified game benchmark specifically designed for MLLMs, featuring diverse semantic environments, randomized layouts, and varying difficulty levels to ensure a generalizable and robust assessment.
- We conducted standard tests on seven MLLMs, and our empirical results highlighted the current capability deficiencies in these models.

## 2 RELATED WORKS

### 2.1 MULTIMODAL LARGE LANGUAGE MODELS

LLMs (Ouyang et al., 2022; Touvron et al., 2023; Chung et al., 2024) have evolved from processing solely text-based inputs to exhibiting multimodal capabilities. This advancement has significantly expanded the applicability of MLLMs in areas such as image description (Liu et al., 2016; Tan et al.,

2024), image reasoning (Ilievski & Feng, 2017; Wang et al., 2024d; Xiao et al., 2024), and visual question answering (VQA) (Gaur et al., 2024; Wang et al., 2024a), bringing us closer to the ultimate goal of AI research: general artificial intelligence (AGI) (Zhong et al., 2024), which aims to develop systems capable of matching or surpassing human-level performance across diverse domains.

## 2.2 MLLM BENCHMARK

Many benchmarks have been introduced to assess the capabilities and performance of MLLMs. However, most of them primarily focus on evaluating MLLMs' ability to process and understand multi-modal data, such as image comprehension and analysis (Li et al., 2023; Xu et al., 2023; Yin et al., 2023; Yu et al., 2023; Fu et al., 2024). In addition to these, some benchmarks specifically evaluate MLLMs' capacity for human-level planning (Chen et al., 2024b), while others focus on improving the models' ability to make embodied decisions (Chen et al., 2023). While these assessments offer valuable insights into MLLMs' performance on individual tasks, there is currently no comprehensive benchmark, akin to GLUE (Wang et al., 2019; Sarlin et al., 2020) for LLMs, that evaluates MLLMs across a diverse range of tasks. Furthermore, existing benchmarks lack a systematic definition of MLLMs' capabilities and do not offer a well-defined classification of assessment tasks, as seen in HELM (Liang et al., 2023) and BIG-Bench (Srivastava et al., 2023).

Using games (Bellemare et al., 2018; Juliani et al., 2019; Samvelyan et al., 2021; Gan et al., 2021; Chevalier-Boisvert et al., 2023) as benchmarking tools offers a unique approach to assessing LLM capabilities. For example, SmartPlay (Wu et al., 2024d) integrates six classic games, including Minecraft (Johnson et al., 2016) and Crafter (Hafner, 2021). It converts game scenarios into text-based descriptions to evaluate core LLM abilities, such as instruction following and error correction. GameBench (Costarelli et al., 2024) utilizes tabletop games to assess reasoning skills. Additionally, concurrent studies have explored the use of board games (Topsakal et al., 2024) and open-ended wargames (Hogan & Brennen, 2024) for benchmarking LLMs. While these benchmarks have significantly advanced LLM development, they primarily focus on text-based evaluations and are not well-suited for MLLMs.

Given that MLLMs are still in the early stages of development, game benchmarks based on existing entertainment-focused games often overwhelm these models with excessive "redundant" details (Li et al., 2024; Zhang et al., 2024a). While complex visual elements such as intricate character designs, background scenery, and narrative dialogue enhance the experience for human players, they do not meaningfully contribute to the models' reasoning and problem-solving processes. Furthermore, many existing games have been documented extensively, which can give an unfair advantage to MLLMs that have been pre-trained on data containing information about these games. As a result, a benchmark that presents well-designed tasks with limited redundant visual complexity, avoids reliance on pre-documented games, and focuses on testing MLLMs' reasoning, problem-solving, and generalization abilities across novel, unseen scenarios would offer a more accurate and meaningful evaluation of their capabilities (Wu et al., 2024a; Zhu et al., 2024).

To address this gap, we develop an original benchmark with well-defined objectives and metrics (see Section 4) specifically designed for evaluating MLLMs. Our framework adopts the Gym interface (Brockman et al., 2016) and introduces a set of game environments that strike an appropriate balance between complexity and semantic richness. These environments are structured around twelve pre-defined tasks (see Section 5) that are carefully designed to target and evaluate MLLMs' core abilities. By incorporating five key abilities derived from human intelligence tests (see Section 3), our benchmark allows for a precise assessment of the strengths and weaknesses of MLLMs, offering a more focused and effective evaluation of their capabilities.

## 3 CAPABILITIES

The cognitive development of human provides essential insights for creating truly flexible and intelligent agents (Wu et al., 2024c; Sumers et al., 2024). By borrowing concepts from Wechsler Intelligence Scales, a well-established framework for assessing children's intelligence, we identify five key abilities crucial for MLLMs.

**Exuction:** All human actions originate from intentions (Searle, 1983). Similarly, MLLMs also needs to transform their understanding of goals into actions to achieve meaningful outcomes. We define execution as the ability of MLLMs to carry out tasks based on their understanding of goals and requirements. Whether navigating a virtual environment, manipulating objects, or interacting with other agents, strong execution is crucial for MLLMs to turn abstract goals into real-world or simulated behaviors, ensuring that their understanding of tasks translates into successful outcomes by effectively carrying out the intended actions.

**Perception Reasoning:** Human cognition relies heavily on the ability to perceive and interpret sensory information, forming the basis for reasoning and decision-making. In traditional reasoning, particularly as it relates to LLMs, the focus is primarily on language, where reasoning processes involve constructing thoughts and inferences articulated through linguistic constructs. In contrast, we define **perception reasoning** as the ability of MLLMs to derive inferences and make decisions directly from visual information, such as images. This process goes beyond merely understanding images; it involves analyzing and reasoning about visual data to form logical conclusions, predict outcomes, and guide actions. Perception reasoning is crucial because many future AGI challenges will require handling multi-modal information (Guan et al., 2024), where reasoning directly from visual inputs and integrating them with other sensory data is key to making comprehensive and accurate decisions.

**Memory:** Humans utilize memory systems to store information and knowledge from past experiences and apply them to current situations (Atkinson & Shiffrin, 1968). In MLLMs, **memory** serves a similar role by allowing them to retain information (Wang et al., 2024c), build context, and improve decision-making over time." Memory allows MLLMs to integrate past observations and learned knowledge into their reasoning process, which is essential for tasks that require an understanding of historical data or sequential patterns.

**Learning:** Learning is a fundamental aspect of human cognition, enabling individuals to acquire new information, rules, and knowledge, and apply them to solve novel problems. Similarly, learning in MLLMs refers to their capacity to absorb new information or rules and effectively utilize this knowledge in decision-making and problem-solving scenarios. A key challenge arises when the new information contradicts or contrasts with previously learned knowledge. For MLLMs, this is particularly difficult when the model has not been fine-tuned on data containing these new rules, as they need to reconcile conflicting knowledge without explicit retraining. Despite these challenges, the ability to adapt to new information is crucial for AGI. Complex real-world problems often evolve over time, requiring MLLMs to learn, adjust, and apply new knowledge dynamically, without relying solely on pre-existing data or frequent fine-tuning, which can be expensive and slow. This adaptability is essential for ensuring that MLLMs remain flexible and robust in diverse, changing environments.

**Planning:** In human intelligence, planning plays a crucial role by allowing individuals to predict future outcomes, devise strategies, and arrange actions in a structured sequence to achieve specific objectives. In the context of MLLMs, **planning** refers to the capability to organize and prioritize tasks, anticipate the consequences of actions, and execute multi-step strategies to address complex problems. This ability extends beyond reactive decision-making, as it requires foresight, where the model must account for long-term goals and carefully weigh the trade-offs between immediate actions and future outcomes.

## 4 GAME MECHANICS

The games in GridAgent has been specifically designed with several mechanisms (see Figure 2) that take into account both the strengths and weaknesses of MLLMs.

**Diverse Semantic Scenes:** In real-world applications, tasks of the same type often vary based on their contextual scenarios. For example, a classification task might involve categorizing items such as "*placing apples and bananas into different baskets*" in a supermarket, or "*placing hamburgers and sandwiches onto separate plates*" in a canteen. While these scenarios differ in context, the

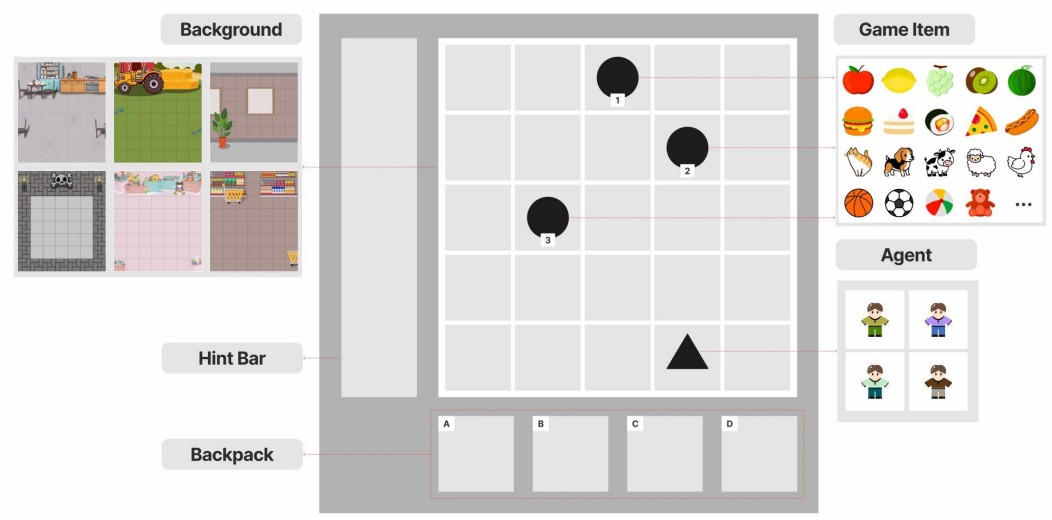

Figure 2: A frame in GridAgent consists of a game scene map, a backpack, and a hint bar. We provide a variety of agent appearances, diverse backgrounds, and a wide range of game items tailored to each semantic scene. The backpack and items are labeled with different alphabetic letters and numbers correspondingly to facilitate identification by MLLMs.

underlying behavioral logic remains the same. To capture these variations, we have designed a range of environments, including supermarkets, canteens, and farms, along with corresponding game items to create immersive, context-rich scenarios. This diversity in semantic scenes is crucial for assessing the generalization and robustness of MLLMs. By testing the model across varying contexts, we can evaluate whether it has developed the targeted abilities and can apply them effectively in different scenarios, rather than relying solely on memorization of pre-training data.

**Randomness:** In addition to diverse semantic scenes, variability in game layouts plays a crucial role in assessing the robustness of MLLMs. While semantic diversity introduces new contexts for each task, randomness focuses on generating different layouts within the same task and scene. Each game is initialized with a randomized arrangement of elements, such as game item placement or agent starting positions, ensuring that no two game instances are identical. This randomness helps reduce the variance in evaluating MLLMs' abilities and ensure more consistent task performance. By introducing layout variability, we can more accurately assess the adaptability and generalization of MLLMs across repeated scenarios with the same underlying goals.

**Backpack:** Current MLLMs often struggle to maintain contextual consistency, especially when handling hidden details not explicitly depicted in visual information. For example, an agent may "*pick up the key*" in one step but fail to recall possessing it in later steps. To address this, we have designed a backpack area as part of the game state, allowing agents to store and access important state information over time. This feature helps MLLMs preserve memory of past actions, enhancing their understanding of environmental changes and optimizing their reasoning abilities."

**High-level Actions:** LLMs are not well-suited for executing atomic actions such as "*go one step forward*" or "*turn left*" in games requiring high operability, partly because they take relatively long to process each command in real-time interactive scenes (Gallotta et al., 2024). In contrast, LLMs are better suited for handling abstract concepts and executing higher-level actions because they are more likely to perform better at reasoning, decision-making, and planning over longer time horizons. Based on this, each task in GridAgent provides LLMs with high-level actions. For example, the agent can directly perform "*pick up the basketball*" rather than navigating step-by-step to its location and interacting with it. This reduction in operational granularity not only alleviates computational load, leading to more efficient processing and faster decision-making, but also ensures the model

Table 1: Twelve games are included in GridAgent, each designed to evaluate at least one of the MLLM's capabilities in Appendix 3. Some tasks provide multiple difficulty levels to challenge more advanced agents. The abbreviations $A, F_o, F_r, T, S.E.$ refer to "*Animal*", "*Food*", "*Fruit*", "*Toy*" and "*Specialized Environment*" respectively.

| Task Type | Required Capacity | Difficulty Levels | Semantic Envs |
|---|---|---|---|
| Classification (CL) | Execution | L1/L2/L3 | $A, F_o, F_r, T$ |
| Selection (SE) | Memory | L1/L2/L3 | $A, F_o, F_r, T$ |
| Decode (DE) | Learning | L1/L2/L3 | $A, F_o, F_r, T$ |
| Maze (MA) | Planning | L1/L2/L3 | $S.E.$ |
| Filling (FI) | Perception Reasoning | L1/L2/L3 | $A, F_o, F_r, T$ |
| Puzzle (PU) | Perception Reasoning (Abstract) | L1/L2/L3 | $S.E.$ |
| Maze* (MA*) | Planning + Memory | L1/L2/L3 | $S.E.$ |
| Decode* (DE*) | Learning + Memory | L1/L2/L3 | $A, F_o, F_r, T$ |
| Sorting (SO) | Learning + Planning | L1/L2/L3 | $A, F_o, F_r$ |
| Filling* (FI*) | Perception Reasoning + Memory | L1/L2/L3 | $A, F_o, F_r, T$ |
| Placement (PL) | Perception Reasoning + Learning | L1/L2/L3 | $A, F_o, F_r, T$ |
| Counting (CO) | Perception Reasoning + Planning | L1/L2/L3 | $F_o, F_r$ |

focuses on actions directly tied to meaningful game outcomes, avoiding the complexity of low-level controls.

**Identification:** Each game scene item and the items in the agent's backpack are assigned unique numerical and alphabetical identifiers. These identifiers enable the MLLMs to associate visual elements with text-based descriptions in high-level actions or goals, such as "*put the object from backpack A into object number 2*." This setup helps evaluate the models ability to link visual information with contextual understanding and execute precise actions. These labels not only optimize information retrieval but also reduce ambiguity in task execution, ensuring that the agent interprets and interacts with the environment accurately. The combination of identification and high-level actions serves as a test of the MLLM's ability to comprehend and reason about images in relation to the game's objectives.

## 5 Tasks in GridAgent

We design a series of tasks to evaluate the individual and composite abilities of MLLMs. Tasks vary in difficulty to challenge both basic and advanced models, while randomized layouts and diverse semantic scenes further enhance the assessment of generalization and robustness. Detailed information specifics for each task can be found in Appendix G.

### 5.1 Single Capacity Task

**Classification (CL):** In this task, the agent is required to place each item into its designated container based on specific instructions, such as "*placing the cherry in the yellow basket*" and "*placing the peach in the blue basket*" (see Figure 1a). It is designed to evaluate the MLLM's **Execution** ability, which involves translating an understanding of goals into effective actions. The agent's performance in this task measures its accuracy in following instructions within a structured environment. The task is designed with varying difficulty levels, where more difficult tasks involve an increased number of items and placement operations. By increasing the complexity of the task, we further assess how well the MLLM can adapt to more demanding scenarios that require precise execution.

**Filling (FI):** During the filling task, the agent will be presented with an image in which a quarter section has been removed, such as "*a goldfish with a missing head*" (see Figure 1i). Then it needs to restore the image by selecting the correct missing piece from a set of distractors in the backpack. This task primarily evaluates the MLLM's **Perception Reasoning** ability, as it requires the agent to develop a holistic understanding of the image and infer the missing part.

**Puzzle (PU):** A target image composed of four puzzle pieces is displayed in the hint bar (see Figure 1j), and the agent needs to assemble the scattered puzzle pieces from its backpack to reconstruct the target image. This task primarily evaluates the MLLM's **Perception Reasoning** ability in abstract visual mode, as it requires the agent to grasp the image's overall structure, which cannot be easily conveyed through language.

**Selection (SE):** In Selection, before the game start, some random items will appear in the left hint bar (see Figure 1e). Once the game starts, these items will be hidden from players. The agent need to select the items appeared in the hint bar before. This task evaluates the MLLM's **Memory** capability by requiring it to remember and recall the items previously shown. As the difficulty increases, the number of items the agent is required to remember also rises.

**Decode (DE):** The agent is provided with a code table, which contains a certain number of association rules between different items (see Figure 1j). The agent needs to first learn these correspondences. When a target item appears in the top left corner of the frame, then the agent is required to select the item that corresponds to the target based on the learned associations. This task primarily assesses the MLLM's **Learning** capability, as it requires the agent to understand the new relationships presented in the code table and apply that knowledge to make an informed decision. As the task difficulty increases, the complexity of the correspondences the agent must remember also increases.

**Maze (MA):** This game is inspired by Procegon (Cobbe et al., 2019), where agent must obtain the diamond in a maze with several locked doors. The agent needs to collect and use the corresponding colored keys to unlock these doors (see Figure 1f). This task primarily evaluates the MLLM's **Planning** ability. Not every door needs to be opened, so the agent should carefully devise a strategy to reach the diamond in as few steps as possible. As the difficulty increases, the number of necessary doors to unlock also rises, and each action taken can significantly influence the agent's subsequent decisions.

## 5.2 COMPOSITE CAPACITY TASK

**Maze* (MA*):** This task follows the same rules as the "*Maze*" described in Section 5.1, with an added challenge. Before the game begins, the agent is shown the location of the diamond, but once the game starts, the diamond is hidden among several treasure chests. To succeed, the agent must correctly open the chest containing the diamond (see Figure 1g). This task primarily assesses the MLLM's **Memory** and **Planning** abilities, as the agent must recall the diamond's location and devise an effective strategy to retrieve it.

**Decode* (DE*):** This task follows the same rules as "*Decode*" in Section 5.1. The agent must additionally remember the relationships indicated in the code table, which will disappear once the game starts. This task primarily evaluates the LLM's abilities in **Memory** and **Learning**, as it requires the agent to retain and utilize the information from the code table to make accurate selections.

**Sorting (SO):** In the sorting task, the agent is presented with a rule that may contradict real-world knowledge. For instance, the agent might be instructed that "*the faster the animal, the heavier it is*". The agent is then expected to correctly rank the animals based on this given rule. This task evaluates the MLLM's **Learning** and **Planning** abilities, as it requires the agent to not only comprehend and integrate novel logic that may conflict with its prior knowledge but also apply it accurately in the sorting process.

**Filling* (FI*):** This task follows the same rule as "*Filling*" in Section 5.1. The agent must additionally remember the target image, which will disappear once the game starts. This task primarily evaluates the LLM's abilities in **Perception Reasoning** and **Memory**, as it necessitates recognizing the overall image structure and recalling specific details to identify the correct piece.

**Placement (PL):** The agent is required to place the item in the opposite position based on the given goal. For instance, if the rule states "*place the toy car on the north side of the toy train*" (see Figure 1h), the agent actually need to place it on the "*south*" side. This task primarily evaluates the MLLM's abilities in **Perception Reasoning** and **Learning**, as it necessitates an understanding of placement rules and the awareness of spatial orientation.

**Counting (CO):** The scene contains several piles of items, with quantities ranging from 1 to 3 (see Figure 1b). At the start of the game, the agent is given a target number and must collect exactly that number of items. As the difficulty increases, more piles of items are introduced and the target number grows, requiring the agent to gather more items from multiple piles. This task primarily evaluates the MLLM's **Perception Reasoning** and **Planning** abilities, focusing on the agent's awareness of item quantities and its strategic decision-making regarding how many items to collect at once.

## 6 EXPERIMENTAL RESULT

We selected seven prominent MLLMs for a comprehensive evaluation, including GPT-4o (OpenAI, 2024), Gemini-1.5-flash (Team, 2024b), Qwen2-VL-7b (Yang et al., 2024), LLaVA-v1.6-Mistral-7b (Liu et al., 2024), Deepseek-v1-7b (Team, 2024a), InternLM-XComposer2-7b (Zhang et al.), Phi-v3.5-Vision (Abdin et al., 2024), and InternVL-Chat-v1.5 (Chen et al., 2024c). To ensure a robust assessment, we first created 500 rounds of games for each task and tested all the above models on the same set. We selected the option with the highest probability from the models' outputs. Aligned with the mainstream MLLM benchmarks (Chen et al., 2024b), we evaluate MLLM's capabilities through multiple-choice questions. This format facilitates the convenient calculation of accuracy as an objective metric. Moreover, it enables us to thoughtfully design incorrect options to control the quality and difficulty of our benchmark. We have included a detailed explanation of the evaluation procedure in Appendix C of the appendix.

### 6.1 QUANTITATIVE ANALYSIS

We observed that most MLLMs performed reliably only at Level 1, with performance dropping significantly, often approaching the random baseline at Levels 2 and 3 (see Appendix B.4). Therefore, we focused on Level 1 results in this section to highlight scenarios where the models demonstrated meaningful capabilities.

**MLLMs still have considerable potential for enhancement as agents.** For humans, these game tasks are relatively easy to complete (see Appendix D). However, as shown in Tables 2 and 3, while GPT-4o's performance is impressive, a significant gap still exists between GPT-4o and the human baseline, with GPT-4o achieving a normalized score above 90% only on MA/MA*/DE* tasks. Most MLLMs, however, performed close to the random baseline across various tasks and still have a long way to go to catch up with GPT-4o.

**Different MLLMs exhibit significant variations in their abilities across different tasks.** Despite the overall suboptimal performance, the data reveal that different MLLMs exhibit distinct strengths. For example, GPT-4o excels in MA (1.00) and MA* (0.99) tasks that require strong planning abilities, while Deepseek surpasses other models in the PL(0.26) task. Additionally, InternVL

Table 2: Comparison of the normalized score (see specific calculation method in Appendix D.3) of different MLLMs on single capacity test: Classification, Selection, Decode, Maze, Filling, Puzzle.

| Level1 | CL | SE | DE | MA | FI | PU |
|---|---|---|---|---|---|---|
| Human | 1.00 | 1.00 | 1.00 | 1.00 | 1.00 | 1.00 |
| GPT-4o | 0.88 | **0.48** | **0.72** | **1.00** | **0.52** | 0.26 |
| Gemini | **0.97** | 0.26 | 0.52 | 0.99 | 0.45 | 0.24 |
| Qwen2 | 0.70 | 0.41 | 0.34 | 0.71 | 0.50 | 0.25 |
| Internvl | 0.61 | 0.32 | 0.26 | 0.90 | 0.38 | 0.22 |
| DeepSeek | 0.49 | 0.24 | 0.35 | 0.89 | 0.41 | **0.27** |
| Phi3.5 | 0.42 | 0.26 | 0.25 | 0.83 | 0.38 | 0.25 |
| Llava | 0.37 | 0.25 | 0.25 | 0.88 | 0.34 | 0.25 |
| InternLM | 0.66 | 0.25 | 0.25 | 0.92 | 0.45 | 0.25 |
| Random | ≈ 0.67 | 0.25 | 0.25 | ≈0.79 | ≈0.27 | 0.25 |

Table 3: Comparison of the normalized score of different MLLMs on multiple capacity test: Maze*, Decode*, Sorting, Filling*, Placement, Counting.

| Level1 | MA* | DE* | SO | FI* | PL | CO |
|---|---|---|---|---|---|---|
| Human | 1.00 | 1.00 | 1.00 | 1.00 | 1.00 | 1.00 |
| GPT-4o | 0.99 | **0.95** | 0.70 | **0.81** | 0.08 | **0.51** |
| Gemini | **1.02** | 0.80 | 0.57 | 0.52 | 0.22 | 0.42 |
| Qwen2 | 0.11 | 0.38 | 0.87 | 0.39 | 0.19 | 0.50 |
| Internvl | 0.46 | 0.27 | **1.16** | 0.31 | 0.15 | 0.49 |
| DeepSeek | 0.17 | 0.26 | 0.69 | 0.27 | **0.26** | 0.42 |
| Phi3.5 | 0.18 | 0.26 | 0.67 | 0.40 | 0.17 | 0.43 |
| Llava | 0.19 | 0.24 | 0.51 | 0.30 | 0.17 | 0.42 |
| InternLM | 0.19 | 0.26 | 0.54 | 0.19 | 0.24 | 0.44 |
| Random | ≈ 0.09 | ≈ 0.25 | 0.60 | ≈ 0.28 | 0.17 | ≈ 0.43 |

demonstrates an ability to quickly understand new rules, achieving a high score (1.16) beyond the human baseline.

**MLLMs have error correction capability and can learn from interactions.** For tasks requiring multiple steps, as shown in Table 3, the agent was initially provided with more steps than necessary to complete the task, but this allowance was removed in the secondary test. Judging from the results, when the temperature of all models was set to 0 and no indication was given that the MLLM had made a wrong choice, the test with the maximum number of steps showed a significantly higher pass rate. This demonstrates that even when an error occurs during interaction with the environment, the MLLM can adjust its subsequent choices based on environmental feedback, rather than repeatedly selecting an invalid action.

**Some MLLMs perform better in the memory counterpart of certain tasks.** GPT-4o and Qwen2-VL exhibit better performance in DE* and FI* tasks compared to their non-memory counterparts, which appears counter-intuitive. However, in our evaluation, most MLLMs acquire memory capabilities by treating previous information as history and concatenating it with the current state as input for inference. Consequently, both the image to be remembered and the image representing the problem state are provided in the prompt. This effectively allows MLLMs to "re-read" the problem, potentially providing an advantage in inference, as observed in Xu et al. (2024a), where repeating a question improves performance. A similar pattern is not seen in SE* and its counterpart, likely due to the confusing information present in SE*'s problem state (e.g., extra fake treasure chests), which undermines the effectiveness of the re-reading strategy.

## 6.2 QUALITATIVE ANALYSIS

**MLLMs' training data lacks image-only perception training.** The PU task showed the lowest success rate among all single-ability tasks, primarily because it relies solely on abstract image perception. The task description only instructs participants to restore the target image without providing clear guidance on the restoration process. This separation between abstract visual content and textual instructions makes the task inherently more difficult.

**It is difficult for MLLMs to learn knowledge contradict ones from training data.** Most MLLMs perform poorly in the Sorting and Placement tasks, where they are required to learn new knowledge that may contradict prior knowledge from their training data. For instance, in the PL task, the best-performing MLLM achieved only 0.27 score, a result that falls within the fluctuation range of random selection. This suggests that MLLMs struggle to adapt to novel or conflicting information, particularly when it requires overriding existing knowledge embedded during training.

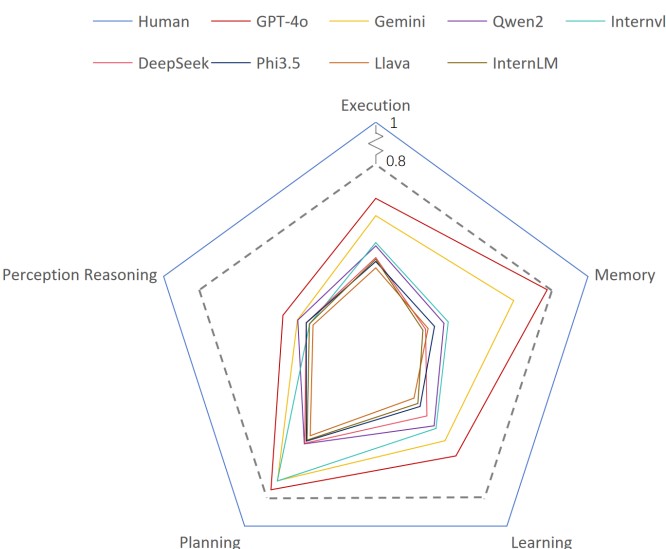

Figure 3: The five-dimensional capability radar map testing against a human baseline, GPT-4o performed the best, while the combined performance of the other models was relatively similar.

### 6.3 RADAR CAPABILITY CHART

To provide further understanding of the individual agent capabilities of MLLM, as discussed in Section 3, we calculated the capability scores and generate a five-dimensional radar chart for each MLLM (see in Figure 3).

For each MLLM, we compute the score for a given capability $c$ by evaluating its performance across all tasks $t_i$ associated with that capability (see in Appendxi F). The score is calculated as:

$$S_c = \frac{1}{n} \sum_{i=1}^{n} R_{t_i}$$

## 7 CONCLUSION

In this work, we introduce GridAgent, a 2D grid-based game environment framework and a unified benchmark for evaluating LLMs as agents. Leveraging the current strengths and weaknesses of MLLMs, our environment provides a rich semantic context, random layouts, high-level actions, and multiple-choice questions. It also offers user-friendly interfaces for developers to easily create game environments tailored to MLLM training.

Building on criteria referenced in human intelligence tests, we propose five novel dimensions to evaluate an MLLM's ability to solve tasks: execution, perception reasoning, memory, learning, and planning. Additionally, our initial release of GridAgent includes twelve goal-oriented tasks designed to assess these capabilities.

We believe GridAgent will offer new datasets and tasks for the MLLM research community, contributing to the continued development and enhancement of AGI.

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
