APPENDIX

# A    BENCHMARK INFORMATION

## A.1    WECHSLER INTELLIGENCE TEST

Our study focuses on the Wechsler Intelligence Scale, specifically the Wechsler Intelligence Scale for Children (WISC), an individually administered test designed for children aged 6 to 16. Originally developed by the American psychologist Dr. David Wechsler, the test undergoes periodic updates and revisions every few decades to maintain its accuracy and relevance. The latest and most widely used version is WISC-V, which measures five primary cognitive domains: Verbal Comprehension Index (VCI), Visual Spatial Index (VSI), Fluid Reasoning Index (FRI), Working Memory Index (WMI), and Processing Speed Index (PSI).

MLLMs are still in the early stages of development. While assessments for adults typically emphasize specialized knowledge or skills, children's tests focus more on fundamental cognitive abilities that underpin general intelligence. By evaluating MLLMs against these foundational abilities, we can more effectively gauge their adaptability and developmental potential, mirroring the stages of human cognitive growth. This approach allows us to gain a comprehensive understanding of MLLMs' strengths and limitations, providing clear guidance for improvements and ultimately contributing to the advancement of AGI.

## A.2    CAPABILITY DESIGN

Wechsler believed that intelligence could be viewed as a combination of various components or abilities that differ in quantity. Here's an explanation of how each capability discussed in the Section 3 connects to the specific cognitive domains measured by the Wechsler Intelligence Test.

**Execution** in MLLMs is connected to the Processing Speed Index (PSI) of the WISC, which assesses the speed and accuracy with which a child completes simple tasks. However, for MLLMs, speed is not the primary focus; we place greater emphasis on accuracy. Therefore, it signifies the model's ability to perform whether simple or complex tasks aligned with specific goals, ensuring that the outcomes are correct and precise.

**Perception Reasoning** in MLLMs aligns with the Visual Spatial Index (VSI) and Fluid Reasoning Index (FRI) of the WISC. While VSI assesses a child's ability to interpret and organize visual information, and FRI evaluates problem-solving and abstract thinking, Perception Reasoning in MLLMs focuses on interpreting input data and making logical inferences, which allows the model to understand context and relationships.

**Memory** in MLLMs corresponds to the Working Memory Index (WMI) of the WISC, which measures a child's capacity to temporarily hold and manipulate information. For MLLMs, Memory involves retaining contextual information over interactions and recalling relevant data when needed. This ensures coherent and contextually appropriate responses.

**Learning** in MLLMs is underpinned by the Verbal Comprehension Index (VCI) from the WISC, which assesses a child's ability to understand and process verbal information. VCI serves as a foundational prerequisite for Learning, as it enables the comprehension necessary for acquiring new knowledge. In designing MLLM capabilities, we considered VCI to ensure that the models have the linguistic understanding required to effectively learn from and adapt to new data.

**Planning** in MLLMs functions similarly to executive processes—cognitive abilities that, although not explicitly isolated in the WISC, are essential for complex tasks. It involves strategizing and sequencing actions to achieve specific objectives while considering potential outcomes. This capacity enables MLLMs to generate coherent, goal-oriented responses or actions, reflecting higher-order thinking skills akin to those that enhance children's performance across various WISC subtests.

# B  EXPERIMENT DETAILS

## B.1  RESOLUTION SETUP

The game environment consists of a $9 \times 9$ grid, where each cell is rendered at a resolution of $64 \times 64$ pixels, resulting in a total resolution of $576 \times 576$ pixels. Within this environment, the central $8 \times 7$ grid area is artistically designed as a background, while a $5 \times 5$ square serves as the active area where the agent can move and interact. The remaining spaces mainly function as decorative elements to provide semantic context. The bottom $8 \times 1$ grid displays the agent's backpack, while the left $9 \times 2$ grid shows hints and task-related information. All item and backpack labels, are rendered at $16 \times 16$ pixels, ensuring clarity and efficient use of space.

## B.2  MODEL PARAMETERS

In this study, we set the temperature parameter of all language models to 0 for all experimental tasks. By doing so, we enforced deterministic behavior, ensuring that the models' outputs were exclusively determined by their learned probability distributions. This configuration minimizes the influence of stochasticity and provides a controlled environment for evaluating model performance.

## B.3  RANDOM BASELINE

To provide a benchmark for comparison, we established a random baseline in which actions were selected entirely at random. The results of the random baseline were obtained either through analytical computation or experimental estimation, depending on the complexity of the task. For simpler tasks, such as SE, DE, FI, PU, DE*, SO, PI* and PL, probabilistic methods were used to compute the expected performance of random actions. For more complex tasks, such as CL, MA, MA* and CO, where analytical solutions are impractical, the random baseline was approximated by running 500 iterations of random experiments. This methodology allows the random baseline to serve as a meaningful point of reference across a diverse range of tasks.

## B.4  RAW DATA

Table 4: Comparison of raw success rate of different MLLMs on Level1 test of Classification, Selection, Decode, Maze, Filling, Puzzle.

| Level1 | CL | SE | DE | MA | FI | PU |
|---|---|---|---|---|---|---|
| Human | 1.00 | 1.00 | 1.00 | 1.00 | 0.94 | 1.00 |
| GPT-4o | 0.88 | **0.48** | **0.72** | **1.00** | **0.49** | 0.26 |
| Gemini | **0.97** | 0.26 | 0.52 | 0.99 | 0.42 | 0.24 |
| Qwen2 | 0.70 | 0.41 | 0.34 | 0.71 | 0.47 | 0.25 |
| Internvl | 0.61 | 0.32 | 0.26 | 0.90 | 0.36 | 0.22 |
| DeepSeek | 0.49 | 0.24 | 0.35 | 0.89 | 0.39 | **0.27** |
| Phi3.5 | 0.42 | 0.26 | 0.25 | 0.83 | 0.36 | 0.25 |
| Llava | 0.37 | 0.25 | 0.25 | 0.88 | 0.32 | 0.25 |
| InternLM | 0.66 | 0.25 | 0.25 | 0.92 | 0.42 | 0.25 |
| Random | ≈0.67 | 0.25 | 0.25 | ≈0.79 | 0.25 | 0.25 |

Table 5: Comparison of raw success rate of different MLLMs on Level1 test of Maze*, Decode*, Sorting, Filling*, Placement, Counting.

| Level1 | MA* | DE* | SO | FI* | PL | CO |
|---|---|---|---|---|---|---|
| Human | 0.94 | 1.00 | 0.83 | 0.89 | 0.72 | 1.00 |
| GPT-4o | 0.93 | **0.95** | 0.58 | **0.73** | 0.06 | **0.51** |
| Gemini | **0.96** | 0.80 | 0.47 | 0.46 | 0.16 | 0.42 |
| Qwen2 | 0.10 | 0.38 | 0.72 | 0.35 | 0.14 | 0.50 |
| Internvl | 0.43 | 0.27 | **0.96** | 0.28 | 0.11 | 0.49 |
| DeepSeek | 0.16 | 0.26 | 0.57 | 0.24 | **0.19** | 0.42 |
| Phi3.5 | 0.17 | 0.26 | 0.56 | 0.36 | 0.12 | 0.43 |
| Llava | 0.18 | 0.24 | 0.42 | 0.27 | 0.12 | 0.42 |
| InternLM | 0.18 | 0.26 | 0.45 | 0.17 | 0.17 | 0.44 |
| Random | ≈0.08 | 0.25 | 0.50 | 0.25 | 0.13 | ≈0.43 |

Table 6: Comparison of raw success rate of different MLLMs on Level2 test of Classification, Selection, Decode, Maze, Filling, Puzzle.

| Level2 | CL | SE | DE | MA | FI | PU |
|---|---|---|---|---|---|---|
| Human | 1.00 | 1.00 | 1.00 | 1.00 | 0.94 | 1.00 |
| GPT-4o | **0.67** | **0.47** | **0.72** | **0.91** | **0.31** | **0.10** |
| Gemini | 0.63 | 0.16 | 0.45 | 0.64 | 0.29 | 0.07 |
| Qwen2 | 0.27 | 0.10 | 0.21 | 0.23 | 0.15 | 0.08 |
| Internvl | 0.18 | 0.10 | 0.19 | 0.62 | 0.12 | 0.08 |
| DeepSeek | 0.14 | 0.06 | 0.20 | 0.34 | 0.16 | 0.09 |
| Phi3.5 | 0.16 | 0.09 | 0.18 | 0.50 | 0.15 | 0.08 |
| Llava | 0.21 | 0.07 | 0.17 | 0.35 | 0.16 | 0.08 |
| InternLM | 0.34 | 0.06 | 0.16 | 0.74 | 0.16 | 0.07 |
| Random | ≈0.36 | ≈0.07 | 0.17 | ≈0.67 | 0.08 | 0.08 |

Table 7: Comparison of raw success rate of different MLLMs on Level2 test of Maze*, Decode*, Sorting, Filling*, Placement, Counting.

| Level2 | MA* | DE* | SO | FI* | PL | CO |
|---|---|---|---|---|---|---|
| Human | 0.83 | 1.00 | 0.72 | 0.89 | 0.67 | 1.00 |
| GPT-4o | **0.65** | **0.98** | 0.13 | **0.27** | 0.17 | 0.48 |
| Gemini | 0.55 | 0.83 | 0.02 | 0.18 | 0.10 | 0.52 |
| Qwen2 | 0.05 | 0.29 | 0.26 | 0.16 | **0.20** | **0.54** |
| Internvl | 0.24 | 0.17 | **0.39** | 0.11 | 0.08 | 0.45 |
| DeepSeek | 0.04 | 0.17 | 0.27 | 0.16 | 0.12 | 0.45 |
| Phi3.5 | 0.14 | 0.20 | 0.34 | 0.13 | 0.09 | 0.47 |
| Llava | 0.03 | 0.16 | 0.19 | 0.17 | 0.12 | 0.42 |
| InternLM | 0.18 | 0.18 | 0.23 | 0.15 | 0.11 | 0.50 |
| Random | ≈0.10 | 0.17 | 0.17 | 0.08 | 0.13 | ≈ 0.48 |

Table 8: Comparison of raw success rate of different MLLMs on Level3 test of Classification, Selection, Decode, Maze, Filling, Puzzle.

| Level3 | CL | SE | DE | MA | FI | PU |
|--------|----|----|----|----|----|-----|
| Human | 0.83 | 0.89 | 1.00 | 1.00 | 0.89 | 0.94 |
| GPT-4o | **0.58** | **0.38** | **0.59** | **0.87** | 0.07 | 0.04 |
| Gemini | 0.41 | 0.05 | 0.35 | 0.53 | **0.11** | 0.04 |
| Qwen2 | 0.16 | 0.02 | 0.16 | 0.44 | 0.08 | 0.03 |
| Internvl | 0.04 | 0.00 | 0.13 | 0.44 | 0.05 | 0.02 |
| DeepSeek | 0.00 | 0.02 | 0.14 | 0.14 | 0.06 | **0.06** |
| Phi3.5 | 0.11 | 0.02 | 0.14 | 0.38 | 0.08 | 0.04 |
| Llava | 0.09 | 0.01 | 0.12 | 0.12 | 0.09 | 0.05 |
| InternLM | 0.15 | 0.01 | 0.12 | 0.47 | 0.04 | **0.06** |
| Random | $\approx 0.23$ | $\approx 0.02$ | 0.13 | $\approx 0.63$ | 0.04 | 0.04 |

Table 9: Comparison of raw success rate of different MLLMs on Level3 test of Maze*, Decode*, Sorting, Filling*, Placement, Counting.

| Level3 | MA* | DE* | SO | FI* | PL | CO |
|--------|-----|-----|----|----|----|-----|
| Human | 0.72 | 0.94 | 0.67 | 0.83 | 0.67 | 1.00 |
| GPT-4o | **0.63** | **0.95** | 0.08 | **0.14** | 0.17 | 0.48 |
| Gemini | 0.39 | 0.83 | 0.04 | 0.09 | 0.10 | 0.52 |
| Qwen2 | 0.08 | 0.24 | 0.08 | 0.08 | **0.20** | **0.54** |
| Internvl | 0.17 | 0.15 | **0.13** | 0.05 | 0.08 | 0.45 |
| DeepSeek | 0.04 | 0.13 | 0.07 | 0.06 | 0.12 | 0.45 |
| Phi3.5 | 0.13 | 0.14 | 0.17 | 0.06 | 0.09 | 0.47 |
| Llava | 0.04 | 0.13 | 0.05 | 0.09 | 0.12 | **0.42** |
| InternLM | 0.16 | 0.13 | 0.04 | 0.07 | 0.11 | 0.50 |
| Random | 0.13 | 0.13 | 0.04 | 0.04 | 0.13 | $\approx 0.48$ |

## C  EVALUATION PROCEDURE

### C.1  CODE STRUCTURE

With a focus on user-friendliness, GridAgent supports for the OpenAI Gym protocol and can be easily installed with pip. The overall code framework for evaluating MLLM is as follows:

---
**Algorithm 1** Model Evaluation
---
**Require:** $env$: gym environment for the task, $model$: MLLM model, $history$: None
1: **while true do**
2:    $prompt \leftarrow$ **GeneratePrompt**$(env)$
3:    $response, history \leftarrow$ **Chat**$(model, prompt, history)$
4:    $action \leftarrow$ **ProcessAnswer**$(response)$
5:    $env$.**step**$(action)$
6:    **if** the task is over **or** reach the maximum number of steps **then**
7:       exit the while loop
8:    **end if**
9: **end while**

---

## C.2  PROMPT DESIGN

To ensure fairness across all MLLMs, we conducted experiments using exactly the same prompts that included both instructions on the game rules and defined goals for each task.

**Game Rule**

*You are currently playing as a character in a 2D game, as depicted in the image.*
*The game rules are as follows:*
*Items in the game scene are all identified by numerical labels, such as 0, 1, 2, 3, etc.*
*The black squares at the bottom of the screen represent your backpack, labeled as A, B, C, and D, each capable of holding only one item.*
*You cannot move to or interact with an item if there is anything between you and that item.*
*Before each step, you will be presented with a series of action options and you should select the letter corresponding to the action you believe is the right choice to achieve the goal.*

**Task Description**

*In this task, your goal is $\langle GOAL \rangle$.*
*Now, game starts!*
*What is the first action you will choose? / What is the next action you will choose?*
*The actions you can choose from are: $\langle ACTIONS \rangle$.*

**Invalid Answer**

*Your answer is invaild, please tell me the action letter you choose. (e.g. A)*

## C.3  ACTION CHOICE

We convert the high-dimensional actions into random options for MLLMs to choose from.

**Example**

The action list is: [*'pick up apple', 'pick up banana', 'pick up orange'*].

The action choice will be: *A) 'pick up orange', B) 'pick up apple', C) 'pick up banana'*.

## C.4  ANSWER DECODE

Even though we repeatedly emphasized in the prompts that the MLLM should respond in the specified format of the options, we were still unable to strictly standardize their response formats. Therefore, we collected a large number of responses to analyze and summarize the characteristics of MLLMs when answering our questions. The final decoding method was determined as follows.

---

**Algorithm 2** Process Answer

---

**Require:** $answer$: string, $actions$: list of strings
  1: **for** each action in $actions$ **do**
  2:   **if** action is in $answer$ **then**
  3:     **return**  index of action
  4:   **end if**
  5: **end for**
  6: $match \leftarrow$ first uppercase letter found in $answer$
  7: **if** $match$ is not None and the index of $match$ in alphabet is less than length of actions **then**
  8:   $index \leftarrow$ index of $match$ in alphabet
  9:   **return**  $index$
 10: **end if**
 11: **return** None

---

**Explanation**

First, we iterate through the list of actions and check whether any action appears in the response generated by the MLLM. If a match is found, the index of the matching action is returned immedi-

ately. If no match is found, we then search for the first single uppercase letter, and check whether its corresponding index falls within the valid range of the action list. If it does, the corresponding index is returned; otherwise, the response is considered invalid.

**Example**

*response*: $A \rightarrow return\ A$

*response*: *I choose action letter B) 'pick up item with label 2'.* $\rightarrow return\ B$

*response*: *Based on all of the information, I choose action C.* $\rightarrow return\ C$

*response*: *I'm sorry, but I can't provide the correct answer as the image does not contain a dog. It appears to be a game with various animals, but none of them are dogs.* $\rightarrow return\ NONE$

*response*: *...?-=\== ..n\n The-1\n\n The-1* $\rightarrow return\ NONE$

# D  HUMAN BASELINE

## D.1  SETUP

The human experiment included 18 college students who had no prior exposure to our GridAgent environment. The participants were divided into three groups of six, with each group assigned to test one of the three difficulty levels. To minimize the potential influence of repeated attempts and accumulated experience, which could artificially inflate success rates, each participant completed only a single iteration of their assigned tasks. This experimental setup closely mirrored the conditions for the MLLM, which also approached each task as a first-time experience.

## D.2  RAW RESULTS

The results of the human test are presented in Table 10. Despite the limited number of human testers (18), the design of the options in this study provides limited freedom for players, resulting in minimal strategy variation among testers. Furthermore, the convergence of strategy choices indicates that the results of this task do not require a large sample size for reliability. Consequently, the sample size of 18 testers is sufficient for establishing a reliable human baseline for comparison.

Table 10: Accuracy rates for tasks of level1/2/3 difficulty in human testing.

| CL1 | CL2 | CL3 | SE1 | SE2 | SE3 | DE1 | DE2 | DE3 | MA1 | MA2 | MA3 |
|------|------|------|------|------|------|------|------|------|------|------|------|
| 1.00 | 1.00 | 0.83 | 1.00 | 1.00 | 0.89 | 1.00 | 1.00 | 1.00 | 1.00 | 1.00 | 1.00 |

| FI1 | FI2 | FI3 | PU1 | PU2 | PU3 | MA*1 | MA*2 | MA*3 | DE*1 | DE*2 | DE*3 |
|------|------|------|------|------|------|------|------|------|------|------|------|
| 0.94 | 0.94 | 0.89 | 1.00 | 1.00 | 0.94 | 0.94 | 0.83 | 0.72 | 1.00 | 1.00 | 0.94 |

| SO1 | SO2 | SO3 | FI*1 | FI*2 | FI*3 | PL1 | PL2 | PL3 | CO1 | CO2 | CO3 |
|------|------|------|------|------|------|------|------|------|------|------|------|
| 0.83 | 0.72 | 0.67 | 0.89 | 0.89 | 0.83 | 0.72 | 0.67 | 0.44 | 1.00 | 1.00 | 1.00 |

From the results, it was observed that some testers made simple mistakes due to carelessness, such as misreading the option letters or misinterpreting item and backpack labels. Additionally, distractions during the test led to testers forgetting previously memorized information.

Surprisingly, the testers did not perform well on tasks requiring learning (only 0.67 in SO3 and 0.44 in PL3). After discussing with them, we identified tha the phrasing of the new rules was somewhat convoluted, making it harder for testers to quickly and accurately internalize the instructions. Thus, some testers struggled to fully grasp the new rules introduced in the tasks, leading to confusion during execution.

These observations highlight natural occurrences, as similar errors such as misreading instructions, forgetting prior information or misunderstanding the task rule are also observed in MLLMs. These errors reflect the challenges inherent in processing and applying new information within a limited context, whether for humans or models. Consequently, such factors must be carefully considered during the evaluation process to ensure a fair and comprehensive comparison of performance.

### D.3 NORMALIZED SCORE

Given the success rate $R_{\text{mllm}}$ of the MLLM, we compare its performance with the human baseline (see in Table 10), and eliminate the impact of absolute values through normalization.

$$S_{\text{mllm}} = 1 - \frac{R_{\text{human}} - R_{\text{mllm}}}{R_{\text{human}}} = \frac{R_{\text{mllm}}}{R_{\text{human}}}$$

## E ADDTIONAL EXPERIMENTS

### E.1 ITEM RECOGNIZE

Table 11: Experiment

| MLLM | Recognize |
| --- | --- |
| GPT-4o | 0.86 |
| Gemini | 0.91 |
| Qwen2-VL-7b | 0.95 |
| Internvl-chat-v1-5 | 0.86 |
| DeepSeek-v1-7b-chat | 0.83 |
| Phi3-5-vision-instruct | 0.61 |
| Llava-v1.6-Mistral-7b | 0.31 |
| InternLM-Xcomposer2-7b-chat | 0.31 |

In the recognize test, we provide the large language model with four different images, which could be of animals, fruits, toys, etc. Then we ask the model, "Tell me which one is $\langle OBJ.NAME \rangle$?" The table above records the accuracy of the responses given by the large language models. From this, we can see that the accuracy of most large language models exceeds 0.6, with some even reaching 0.8 or 0.9. This suggests that most large language models are capable of recognizing various items in the game scene.

However, we also noticed that a few large language models performed poorly in this test. For instance, Llava-v1.6-Mistral-7b's accuracy was relatively low. By analyzing the responses from the language model, we found that many of Llava-v1.6-Mistral-7b's answers were considered invalid(see Section C.4), which explains its low accuracy.(InternLM-Xcomposer2-7b-chat is the same.) So generally speaking, most large language models possess a strong ability to recognize objects.

### E.2 RESOLUTION TEST

In our initial setup, each grid cell had a resolution of 64x64. To further enhance our analysis, we conducted additional experiments by testing grid cells at resolutions of 0.5x(32*32), 0.75x(48*48), 1.25x(80*80), and 1.5x(96*96) the original size. These supplementary tests were designed to explore the impact of varying grid resolutions on our results, providing deeper insights into how resolution influences performance and outcomes in our experiments.

Table 12: The results of the success rate of all models in Resolution Test on FI.

| Resolution | 32*32 | 48*48 | 64*64 | 80*80 | 96*96 |
|---|---|---|---|---|---|
| GPT-4o | 0.51 | 0.46 | 0.49 | **0.62** | 0.48 |
| Gemini | 0.38 | 0.42 | 0.42 | 0.41 | **0.44** |
| Qwen2 | 0.29 | 0.40 | **0.47** | 0.43 | 0.45 |
| Internvl | 0.34 | 0.31 | 0.36 | **0.42** | 0.34 |
| DeepSeek | **0.45** | 0.43 | 0.39 | 0.39 | **0.45** |
| Phi3.5 | 0.32 | 0.31 | 0.36 | **0.43** | 0.32 |
| Llava | 0.33 | 0.37 | 0.32 | **0.44** | 0.38 |
| InternLM | 0.32 | 0.34 | **0.42** | 0.37 | 0.35 |

### E.3 EXAMPLE TEST

We incorporated a fully correct solution process for the tasks PI, SO, and PL and used it as the initial prompt provided to the MLLMs. This approach was aimed at investigating whether including an accurate solution as a guiding example would influence the model's performance. By doing so, we sought to assess the extent to which the model leverages such examples to improve its problem-solving capabilities and overall response quality.

Table 13: The results of the success rate of all models in Example Test.

| Example | FI | | SO | | PL | |
|---|---|---|---|---|---|---|
| | Origin | New | Origin | New | Origin | New |
| GPT-4o | 0.49 | 0.59↑ | 0.58 | 0.76↑ | 0.06 | 0.31↑ |
| Gemini | 0.42 | 0.56↑ | 0.47 | 0.69↑ | 0.16 | 0.17↑ |
| Qwen2 | 0.47 | 0.36↓ | 0.72 | 0.91↑ | 0.14 | 0.11↓ |
| Internvl | 0.36 | 0.41↑ | 0.96 | 0.63↓ | 0.11 | 0.17↑ |
| DeepSeek | 0.39 | 0.41↑ | 0.57 | 0.98↑ | 0.19 | 0.10↓ |
| Phi3.5 | 0.36 | 0.44↑ | 0.56 | 0.91↑ | 0.12 | 0.10↓ |
| Llava | 0.32 | 0.46↑ | 0.42 | 0.84↑ | 0.12 | 0.17↑ |
| InternLM | 0.42 | 0.36↓ | 0.45 | 0.72↑ | 0.17 | 0.11↓ |

**Example of Filling Task**

*To help you better understand the game rules, let's walk through a correct example. At the start of the game, you will see the following setup: the target image is displayed on the left side, while the incomplete image is shown inside a frame on the right side. Your goal is to make the framed image on the right identical to the target image by selecting and placing the correct puzzle pieces in the specified positions. In this example, the target image on the left shows a sheep, while the framed image on the right is missing the lower-left corner, specifically the front feet of the sheep. By examining the four available puzzle pieces, we can determine that piece C matches the missing feet. In addition to shape, the color of the piece further confirms that piece C is correct, as its color matches that of the target image, unlike the other options. Thus, the correct choice is: Place piece from backpack C in grid I.*

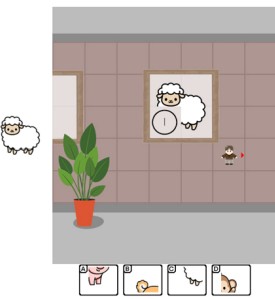

Figure 4: "*Example image for Filling Task*"

**Example of Placement Task**

*To help you better understand the game rule, let's walk through a correct example. At the start of the game, you will see the following setup: Around the hamburger, there are eight grids marked with Roman numerals from I to VIII. Then, you will get the rule: 'Place the item in the opposite direction, such as if it requires you to put it on the east side, you need to place it on the west side.' After you understand the rule, it needs you to 'Place the chicken leg on the east side of the hamburger.' Since the new rule tells that you need to put the item in the opposite direction, the opposite direction of east is west. Therefore, you need to place the chicken leg on the west side of the hamburger, that is label VII. Thus, the correct choice is: place chicken leg at grid VII.*

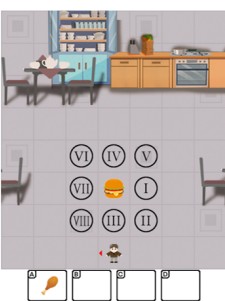

Figure 5: "*Example image for Placement Task*"

**Example of Sorting Task**

*To help you better understand the game rules, let's walk through a correct example. At the start of the game, you will see the following setup: There are two animals in your backpack, a mouse in backpack A and a pig in backpack B. Then, you will get the rule: 'In this world, the lighter the animal is, the slower it is.' After understanding the rule, the task requires you to 'Rank the animals in the backpack from slow to fast by speed in position I, II.' According to the rule, 'the lighter the animal is, the slower it is,' you should know that a mouse is lighter than a pig, so the mouse runs slower than the pig. Although this contradicts common sense, we need to follow the new rules. Thus, the slower animal, the mouse in backpack A, should be placed at grid I, and the faster animal, the pig in backpack B, should be placed at grid II. Therefore, the first correct choice is: 'place animal from backpack A at grid I'.*

Figure 6: *"Example image for Sorting Task"*

### E.4    MODEL SIZE TEST

To evaluate the impact of model size on task performance, we conducted additional experiments using the Qwen2-VL-72b model and compared its results against the Qwen2-VL-7b model on all Level1 tasks.

Table 14: The results of the success rate of two models in Model Size Test on all Level1 tasks.

| Qwen | CL | SE | DE | MA | FI | PU | MA* | DE* | SO | FI* | PL | CO |
|------|------|------|------|------|------|------|------|------|------|------|------|------|
| 72b | **0.79** | 0.24 | 0.27 | **0.98** | **0.52** | 0.21 | **0.31** | **0.38** | 0.71 | **0.40** | 0.06 | **0.59** |
| 7b | 0.70 | **0.41** | **0.34** | 0.71 | 0.47 | **0.25** | 0.10 | **0.38** | **0.72** | 0.35 | **0.14** | 0.50 |

## F    CAPABILITY SCORE

The score for each capability is calculated by the following games:

- Execution: All Games

- Perception Reasoning: FI, PU, FI*, PL, CO

- Memory: SE, MA*, DE*, FI*

- Learning: DE, DE*, SO, PL

- Planning: MA, MA*, SO, CO

For each MLLM, we compute the score for a given capability $c$ by evaluating its performance across all tasks $t_i$ associated with that capability. The score is calculated as:

$$S_c = \frac{1}{n} \sum_{i=1}^{n} R_{t_i}$$

where $S_c$ represents the MLLM's score for capability $c$, $n$ is the total number of tasks related to $c$, and $R_{t_i}$ denotes the success rate of the MLLM on task $t_i$.

Table 15: The Capability Score of each models.

| | Execution | Memory | Learning | Planning | Perception Reasoning |
|---|---|---|---|---|---|
| Human Baseline | 1.00 | 1.00 | 1.00 | 1.00 | 1.00 |
| GPT-4o | **0.70** | **0.81** | **0.61** | **0.80** | **0.44** |
| Gemini | 0.58 | 0.65 | 0.53 | 0.75 | 0.37 |
| Qwen2 | 0.45 | 0.32 | 0.45 | 0.55 | 0.37 |
| Internvl | 0.46 | 0.34 | 0.46 | 0.75 | 0.31 |
| DeepSeek | 0.39 | 0.24 | 0.39 | 0.54 | 0.33 |
| Phi3.5 | 0.38 | 0.28 | 0.34 | 0.53 | 0.33 |
| Llava | 0.35 | 0.25 | 0.29 | 0.50 | 0.30 |
| InternLM | 0.39 | 0.22 | 0.32 | 0.52 | 0.31 |

# G TASK INFORMATION

## G.1 CLASSIFICATION

**Introduction**

This task requires agent to place items into designated containers according to given instructions.

**Goal**

*Place $\langle ITEM_1 \rangle$ in $\langle CONT_1 \rangle$ and $\langle ITEM_2 \rangle$ in $\langle CONT_2 \rangle$ respectively.*

**Actions**

*pick up item with label $\langle CONT.ID \rangle$*
*put the item from backpack $\langle BAG.ID \rangle$ into the basket with label $\langle CONT.ID \rangle$*

**Difficulty Level**

*Level1: The agent needs to put two different kinds of items into the corresponding basket.*
*Level2: The agent needs to put four items of two different kinds into the corresponding basket.*
*Level3: The agent needs to put four items of two different kinds into the corresponding basket.*

**Example**

This is an example of task Classification Level1 successfully completed by Gpt4o.

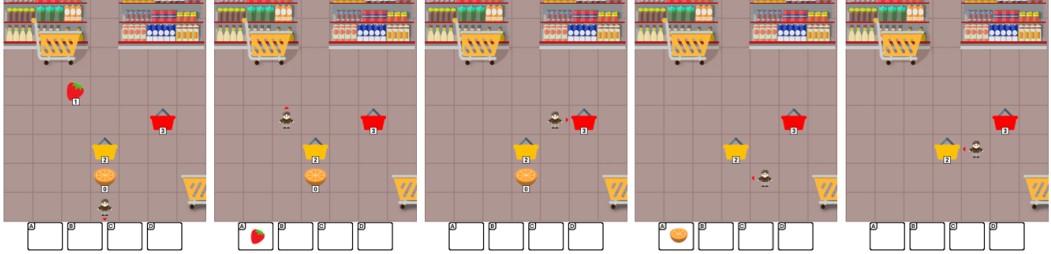

Figure 7: *"The progress of Gpt4o finishing Classification Level1"*

- Step1
  - Action List
    * A) 'pick up item with label 1'
    * B) 'pick up item with label 0'
  - Model's Response
    * **A) 'pick up item with label 1'**
- Step2

- Action List
  * A) 'put the item from backpack A into the basket with label 2'
  * B) 'put the item from backpack A into the basket with label 3'
  * C) 'pick up item with label 0'
- Model's Response
  * **B) 'put the item from backpack A into the basket with label 3'**
- Step3
  - Action List
    * A) 'pick up item with label 0'
  - Model's Response
    * **A) 'pick up item with label 0'**
- Step4
  - Action List
    * A) 'put the item from backpack A into the basket with label 3'
    * B) 'put the item from backpack A into the basket with label 2'
  - Model's Response
    * **B) 'put the item from backpack A into the basket with label 2'**

### G.2 COUNTING

**Introduction**

This task requires agent to pick up a certain number of items.

**Goal**

*Colletct $\langle N \rangle \langle ITEM \rangle$.*

**Actions**

*pick up $\langle ITEM \rangle$ with label $\langle ITEM.ID \rangle$*

**Difficulty Level**

*Level1: The agent needs to collect 1-3 items.*
*Level2: The agent needs to collect 4-6 items.*
*Level3: The agent needs to collect 7-10 items.*

**Example**

This is an example of task Counting Level1 successfully completed by Gpt4o.

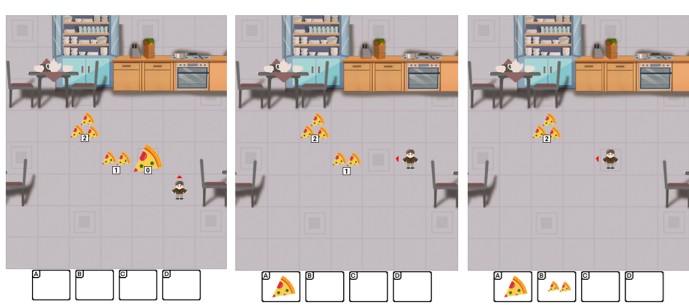

Figure 8: "*The progress of Gpt4o finishing Counting Level1*"

- Step1
  - Action List
    * A) 'pick up pizza with label 1'

* B) 'pick up pizza with label 2'
* C) 'pick up pizza with label 0'
    – Model's Response
        * **C) 'pick up pizza with label 0'**
* Step2
    – Action List
        * A) 'pick up pizza with label 2'
        * B) 'pick up pizza with label 1'
    – Model's Response
        * **B) 'pick up pizza with label 1'**

### G.3 SELECTION

**Introduction**

This task requires agent to memorize the items in the hint bar and collect all of them in the game frame.

**Goal**

*Memorize the item(s) on the left side of the game scene, then based on your memory, select the item(s) that have appeared before.*

**Actions**

*choose ⟨ITEM⟩ with label ⟨ITEM.ID⟩*

**Difficulty Level**

*Level1: The agent needs to remember one item in the hint bar and choose the right thing.*
*Level2: The agent needs to remember two items in the hint bar and choose the right things.*
*Level3: The agent needs to remember three items in the hint bar and choose the right things.*

**Example**

This is an example of task Selection Level1 successfully completed by Gpt4o.

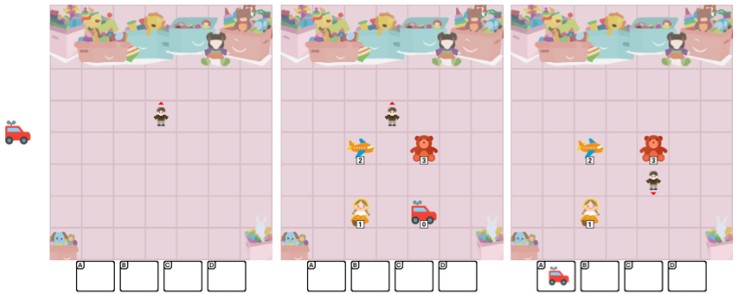

Figure 9: *"The progress of Gpt4o finishing Selection Level1"*

* Step1
    – Action List
        * A) 'skip'
    – Model's Response
        * **A) 'skip'**
* Step2
    – Action List
        * A) 'choose toy with label 3'

                * B) 'choose toy with label 1'
                * C) 'choose toy with label 0'
                * D) 'choose toy with label 2'
        – Model's Response
                * **C) 'choose toy with label 0'**

## G.4 DECODE

**Introduction**

This task requires agent to learn the association rules from a provided code table and using these learned correspondences to select the correct item.

**Goal**

*Learn the association rules between items from the code table on the left of the game scene. Then you need to select the corresponding item to the target item inside the black sqaure on the top of the code table.*

**Actions**

*choose $\langle ITEM \rangle$ with label $\langle ITEM.ID \rangle$*

**Difficulty Level**

*Level1: One set of correspondences is displayed in the hint bar on the left.*
*Level2: Two sets of correspondences are displayed in the hint bar on the left.*
*Level3: Three sets of correspondences are displayed in the hint bar on the left.*

**Example**

This is an example of task Decode Level1 successfully completed by Gpt4o.

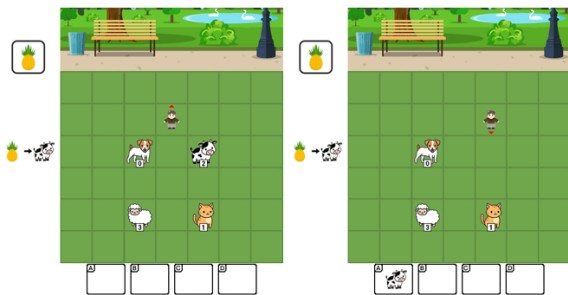

Figure 10: *"The progress of Gpt4o finishing Decode Level1"*

• Step1
    – Action List
        * A) 'choose item with label 3'
        * B) 'choose item with label 2'
        * C) 'choose item with label 0'
        * D) 'choose item with label 1'
    – Model's Response
        * **To achieve the goal, we need to find the item associated with the target item (Pineapple) according to the code table. The code table shows that the cow (item 2) is associated with the Pineapple. Therefore, the first action letter I will choose is B) 'choose item with label 2'.**

## G.5 DECODE*

**Introduction**

This task requires agent not only to learn but to remember the association rules from a provided code table and using these learned correspondences to select the correct item.

**Goal**

*Memorize the association rules between items from the code table on the left of the game scene. Then, based on your memory, you need to select the corresponding item to the target item inside the black sqaure on the left side of the game scene.*

**Actions**

*choose* $\langle ITEM \rangle$ *with label* $\langle ITEM.ID \rangle$

**Difficulty Level**

*Level1: One set of correspondences is displayed in the hint bar on the left.*
*Level2: Two sets of correspondences are displayed in the hint bar on the left.*
*Level3: Three sets of correspondences are displayed in the hint bar on the left.*

**Example**

This is an example of task Decode* Level1 successfully completed by Gpt4o.

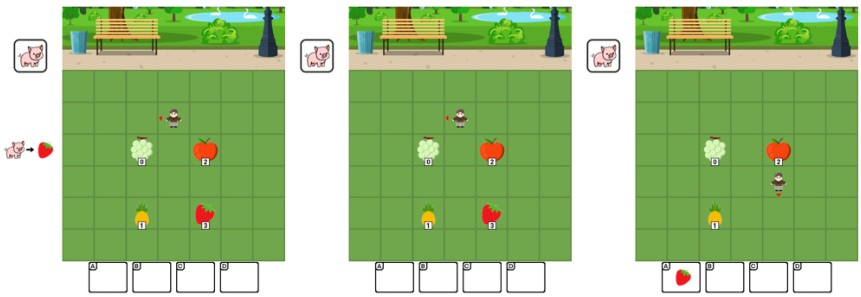

Figure 11: *"The progress of Gpt4o finishing Decode* Level1"*

- Step1
    - Action List
        * A) 'skip'
    - Model's Response
        * **I choose action letter A) 'skip'.**
- Step2
    - Action List
        * A) 'choose item with label 0'
        * B) 'choose item with label 2'
        * C) 'choose item with label 1'
        * D) 'choose item with label 3'
    - Model's Response
        * **I choose action letter D) 'choose item with label 3'.**

G.6   PUZZLE

**Introduction**

This task requires agent to reconstruct a abstract target image by assembling scattered puzzle pieces from its backpack based on the visual reference provided in the hint bar.

**Goal**

*Use the four puzzle pieces from your backpack to form the shape shown on the left of the game scene, pieces need to be properly placed in grid I, II, III, IV.*

**Actions**

*put the puzzle piece in backpack $\langle BAG.ID \rangle$ at grid $\langle GRID.ID \rangle$*

**Difficulty Level**

*Level1: The agent needs to select one picture piece to fill in the missing part.*
*Level2: The agent needs to select two picture pieces to fill in the missing part.*
*Level3: The agent needs to select three picture pieces to fill in the missing part.*

**Example**

This is an example of task Puzzle Level1 successfully completed by Gpt4o.

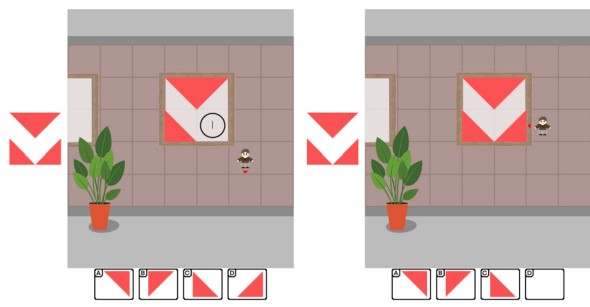

Figure 12: *"The progress of Gpt4o finishing Puzzle Level1"*

- Step1
  - Action List
    * A) 'place piece in backpack D at grid I'
    * B) 'place piece in backpack C at grid I'
    * C) 'place piece in backpack A at grid I'
    * D) 'place piece in backpack B at grid I'
  - Model's Response
    * **I choose action letter \*\*A\*\*: 'place piece in backpack D at grid I'.**

### G.7 FILLING

**Introduction**

This task requires agent to reconstruct a figurative animal by assembling scattered pieces from its backpack based on the visual reference provided in the hint bar.

**Goal**

*See the diagram on the left side of the game scene. Then, select a picture piece from the backpack to fill in the missing part of the picture frame on the wall.*

**Actions**

*place piece in backpack $\langle BAG.ID \rangle$ at grid $\langle GRID.ID \rangle$*

**Difficulty Level**

*Level1: The agent needs to select one picture piece to fill in the missing part.*
*Level2: The agent needs to select two picture pieces to fill in the missing part.*
*Level3: The agent needs to select three picture pieces to fill in the missing part.*

**Example**

This is an example of task Filling Level1 successfully completed by Gpt4o.

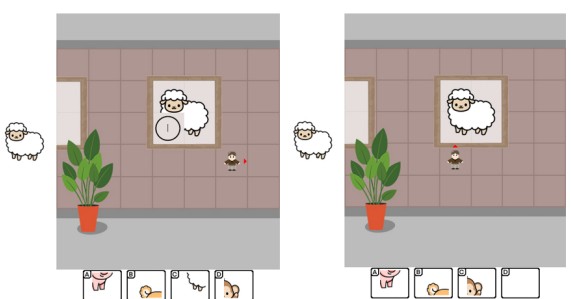

Figure 13: *"The progress of Gpt4o finishing Filling Level1"*

- Step1
    - Action List
        * A) 'place piece in backpack D at grid I'
        * B) 'place piece in backpack B at grid I'
        * C) 'place piece in backpack C at grid I'
        * D) 'place piece in backpack A at grid I'
    - Model's Response
        * **I choose action C) 'place piece in backpack C at grid I'.**

## G.8   FILLING*

**Introduction**

This task requires agent to remember a figurative animal in the hint bar and reconstruct it by assembling scattered pieces from its backpack.

**Goal**

*Memorize the diagram on the left side of the game scene. Then, based on your memory, select a picture piece from the backpack to fill in the missing part of the picture frame on the wall.*

**Actions**

*place piece in backpack $\langle BAG.ID \rangle$ at grid $\langle GRID.ID \rangle$*

**Difficulty Level**

*Level1: The agent needs to select one picture piece to fill in the missing part.*
*Level2: The agent needs to select two picture pieces to fill in the missing part.*
*Level3: The agent needs to select three picture pieces to fill in the missing part.*

**Example**

This is an example of task Filling* Level1 successfully completed by Gpt4o.

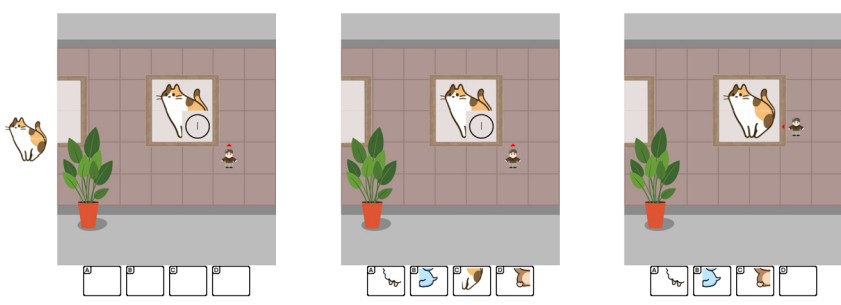

Figure 14: *"The progress of Gpt4o finishing Filling* Level1"*

- Step1
  - Action List
    * A) 'skip'
  - Model's Response
    * **A) 'skip'.**
- Step2
  - Action List
    * A) 'place piece in backpack D at grid I'
    * B) 'place piece in backpack A at grid I'
    * C) 'place piece in backpack C at grid I'
    * D) 'place piece in backpack B at grid I'
  - Model's Response
    * **C) 'place piece in backpack C at grid I'.**

## G.9   MAZE

**Introduction**

This task requires agent to use the keys to unlock corresponding doors to get the diamond.

**Goal**

*Obtain the diamond in the dungeon. When your path is blocked by a door, you can use a key of the same color to unlock it.*

**Actions**

*obtain object with label ⟨ITEM.ID⟩*
*use the key in backpack ⟨KEY.ID⟩ to unlock door with label ⟨DOOR.ID⟩*

**Difficulty Level**

*Level1: The agent needs to open no more than one door to get to the diamond.*
*Level2: The agent needs to open no more than two doors to get to the diamond.*
*Level3: The agent needs to open no more than three doors to get to the diamond.*

**Example**

This is an example of task Maze Level1 successfully completed by Gpt4o.

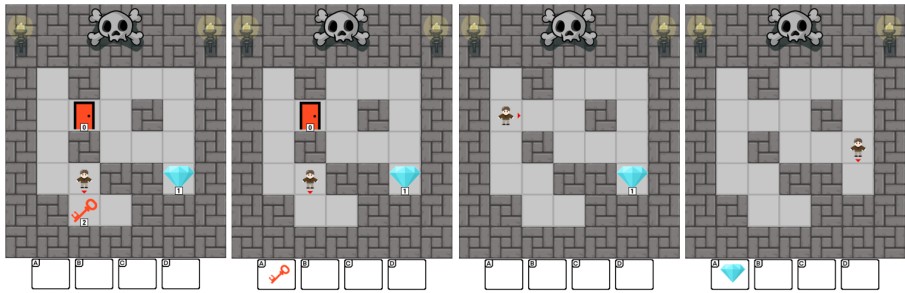

Figure 15: "*The progress of Gpt4o finishing Maze Level1*"

- Step1
  - Action List
    * A) 'obtain object with label 1'
    * B) 'obtain object with label 2'
  - Model's Response
    * **I choose action B) 'obtain object with label 2'.**

- Step2
  - Action List
    * A) 'use the key in backpack A to unlock door with label 0'
    * B) 'obtain object with label 1'
  - Model's Response
    * **I choose action A) 'use the key in backpack A to unlock door with label 0'.**
- Step3
  - Action List
    * A) 'obtain object with label 1'
  - Model's Response
    * **I choose action A) 'obtain object with label 1'.**

## G.10   MAZE*

**Introduction**

This task requires agent to remember the location of the diamond and use the keys to unlock corresponding doors to get the diamond.

**Goal**

*Memorize the location of the diamond in the dungeon. Then, based on your memory, navigate to the correct treasure chest and obtain the diamond from it. When your path is blocked by a door, you can use a key of the same color to unlock it.*

**Actions**

*obtain object with label $\langle ITEM.ID \rangle$*
*use the key in backpack $\langle KEY.ID \rangle$ to unlock door with label $\langle DOOR.ID \rangle$*

**Difficulty Level**

*Level1: The agent needs to open no more than one door to get to the diamond.*
*Level2: The agent needs to open no more than two doors to get to the diamond.*
*Level3: The agent needs to open no more than three doors to get to the diamond.*

**Example**

This is an example of task Maze* Level1 successfully completed by Gpt4o.

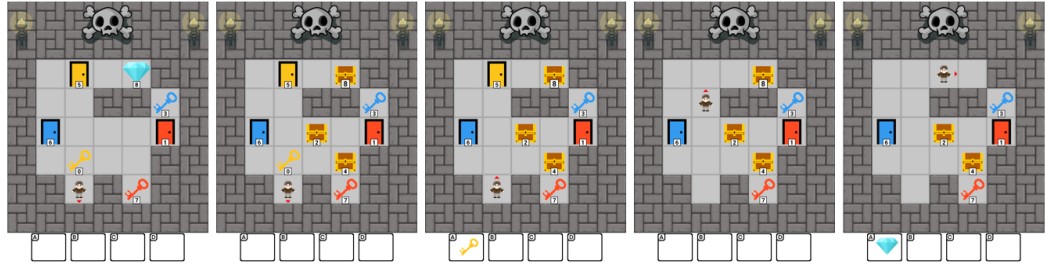

Figure 16: "*The progress of Gpt4o finishing Maze* Level1*"

- Step1
  - Action List
    * A) 'skip'
  - Model's Response
    * **A) 'skip'.**
- Step2

- – Action List

  - * A) 'obtain object with label 0'
  - * B) 'obtain object with label 1'
  - * C) 'obtain object with label 2'
  - * D) 'obtain object with label 4'

- – Model's Response

  - * **C) 'obtain object with label 2'**

- Step3

  - – Action List

    - * A) 'use the key in backpack A to unlock door with label 3'
    - * B) 'obtain object with label 1'
    - * C) 'obtain object with label 0'
    - * D) 'obtain object with label 4'

  - – Model's Response

    - * **A) 'use the key in bag A to unlock door with label 3'.**

- Step4

  - – Action List

    - * A) 'obtain object with label 1'
    - * B) 'obtain object with label 0'
    - * C) 'obtain object with label 4'

  - – Model's Response

    - * **A) 'obtain object with label 1'.**

### G.11 SORTING

**Introduction**

This task requires the agent to sort items based on a provided rule, even if the rule contradicts real-world knowledge.

**Goal**

*In this world, $\langle RULE \rangle$. Rank the $\langle TYPE \rangle$ in the backpack by $\langle PROPERTY \rangle$ in position I, II, III.*

**Actions**

*place $\langle TYPE \rangle$ from bagpack $\langle BAG.ID \rangle$ at grid $\langle GRID.ID \rangle$*

**Difficulty Level**

*Level1: The agent needs to learn the new rule and sort the two things in corresponding order.*
*Level2: The agent needs to learn the new rule and sort the three things in corresponding order.*
*Level3: The agent needs to learn the new rule and sort the four things in corresponding order.*

**Example**

This is an example of task Sorting Level1 successfully completed by IntrenVL.

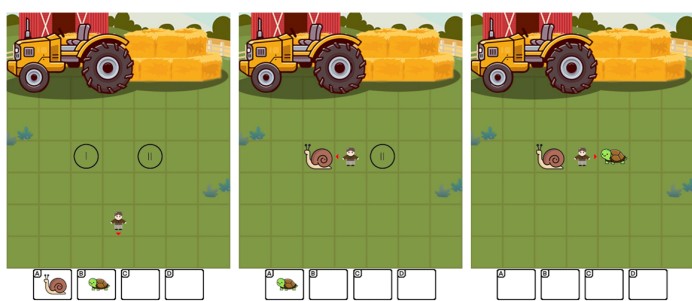

Figure 17: *"The progress of Internvl finishing Sorting Level1"*

• Step1

    – Action List

        ∗ A) 'place animal from backpack B at grid I'

        ∗ B) 'place animal from backpack A at grid II'

        ∗ C) 'place animal from backpack A at grid I'

        ∗ D) 'place animal from backpack B at grid II'

    – Model's Response

        ∗ **C) 'place animal from backpack A at grid I'**

• Step2

    – Action List

        ∗ A) 'place animal from backpack A at grid II'

    – Model's Response

        ∗ **A) 'place animal from backpack A at grid II'**

### G.12 PLACEMENT

**Introduction**

The task requires the agent to interpret the placement rules and place the item in the specified location.

**Goal**

*Place the item in the opposite direction, such as if it is on the east side, you need to place it on the west side. Please place the $\langle ITEM_1 \rangle$ on the $\langle ORIENTATION \rangle$ side of the $\langle ITRM_2 \rangle$.*

**Actions**

*place $\langle ITEM \rangle$ at grid $\langle GRID.ID \rangle$*

**Difficulty Level**

*Level1: The agent needs to put one item in opposite positions of the given position.*
*Level2: The agent needs to place an item either clockwise or counterclockwise from a given location.*
*Level3: The agent needs to place an item clockwise or counterclockwise in the opposite direction of a given location.*

**Example**

This is an example of task Placement Level1 successfully completed by Deepseek.

Figure 18: "*The progress of Deepseek finishing Placement Level1*"

- Step1
  - Action List
    * A) 'place elephant at grid I'
    * B) 'place elephant at grid II'
    * C) 'place elephant at grid V'
    * D) 'place elephant at grid VI'
    * E) 'place elephant at grid III'
    * F) 'place elephant at grid IV'
    * G) 'place elephant at grid VII'
    * H) 'place elephant at grid VIII'
  - Model's Response
    * **G) 'place elephant at grid VII'**