# OpenReview forum: "GridAgent: A 2D Grid-Based Game Framework And Benchmark For Multimodal Large Language Models"
_ICLR.cc/2025/Conference — Submitted to ICLR 2025_

### Official Review · Reviewer_SeHM · 2024-10-31

**Soundness:** 3
**Presentation:** 2
**Contribution:** 3
**Rating:** 6
**Confidence:** 3

**Summary:**

The paper introduces GridAgent, a 2D grid-based benchmark framework designed to evaluate Multimodal Large Language Models (MLLMs) across five core capabilities: execution, perception reasoning, memory, learning, and planning. This is accomplished through twelve distinct game tasks. The results indicate that while models like GPT-4o perform well in specific tasks, none achieve human-level performance across all categories.

The article currently still has some obvious issues and writing structure that need to be addressed. Nonetheless, this contribution is significant, and with further clarification during the rebuttal phase, it could be accepted for the conference.

**Strengths:**

- The benchmark is well-structured, effectively targeting key competencies in MLLMs and providing a robust evaluation framework.
- The introduction of randomized game layouts enhances the test's generalization and robustness by minimizing overfitting to training data.
- The paper presents detailed empirical evaluations of multiple MLLMs, offering clear comparative insights.
- The use of diverse tasks ensures a comprehensive assessment of the models’ abilities, extending beyond simple task-solving to encompass more complex multimodal reasoning, while minimizing overlap with pre-training data, thus enhancing validity.

**Weaknesses:**

- There is a limited discussion regarding the selection of specific game tasks and their comprehensive coverage of the intended capabilities.
- For some tasks, such as Puzzle and Sorting, model performance results close to random, raising questions about the alignment of task complexity with the current capabilities of models.
- Although the paper mentions the randomness in game layouts to avoid overfitting, it lacks sufficient analysis on how varying levels of task complexity affect model performance or whether simpler versions of the benchmark could serve as a more effective baseline. This introduces potential biases due to the artificial nature of the tasks.
- The comparison between MLLMs and human benchmarks appears overly simplistic, lacking a detailed exploration of potential improvements to models in order to bridge the performance gap.

**Questions:**

- The explanation of certain tasks, such as Perception Reasoning, lacks clarity regarding how performance metrics are derived and how these align with human cognition benchmarks.
- Section 5 exhibits some structural confusion, indicating significant issues in the writing structure of the Methodology section. The structure could be better organized to flow logically into the methodology and experimental analysis.
- The current comparative approach between MLLMs and humans is somewhat simplistic; exploring alternative comparison methods could provide deeper insights.

---

> ### Author Response · Authors · 2024-11-23
> **Response to Reviewer SeHM 1**
>
> **Thank you for recognizing the significance of our work.** We sincerely appreciate your thoughtful feedback and have provided detailed responses to your questions and concerns below:
>
> W1 **(Limited Discussion)** Our benchmark is fundamentally inspired by the Wechsler Intelligence Scale, which identifies five core cognitive abilities. Based on these dimensions, we then carefully designed the tasks to evaluate these abilities in a systematic and comprehensive manner.
>
> Single-ability Tasks: For each of the five abilities, we crafted a dedicated task that isolates the specific skill being evaluated. For example:
>
> - Execution is assessed in CL task, which requires the agent to translate explicit instructions into actions.
> - Memory is evaluated in SE task, where agents recall and identify previously seen items.
> - Learning is tested in DE task, where agents must learn and apply new association rules.
> - Planning is examined in MA task, where agents devise efficient strategies to reach a goal.
> - Perception reasoning is tested in FI task, requiring agents to infer missing visual elements from context.
>
> Composite-ability Tasks: To reflect the complexity of real-world scenarios and evaluate higher-order capabilities, we incorporated tasks that combine two abilities. For instance:
>
> - MA* task integrate Planning and Memory by requiring the agent to recall hidden information while optimizing its route.
> - CO task merge Perception Reasoning and Planning, testing the agent’s ability to process quantitative information while executing strategic decisions to meet specific goals.
>
> By combining single ability and composite ability tasks with multiple difficulty levels, our benchmark effectively captures the strengths and weaknesses of MLLMs across multiple dimensions. You can find a more detailed explanation in Section 5. If you want to know how we designed these capabilities, you can refer to Section A.2 of the appendix.
>
> W2 **(Task Complexity)** & W3.1 **(Complexity Effect)** We addressed the details of the difficulty levels in **C1 of the Global Response**. Please refer to that section for a comprehensive explanation.
>
> W3.2 **(Simpler Version as Baseline)** Our primary goal in this work is to evaluate MLLM agents with human-level capabilities.  Therefore, our comparisons are consistently grounded in human performance.  The scoring system we propose for MLLMs is normalized using human scores, ensuring that the evaluation remains relevant and provides a meaningful reference point, regardless of task simplicity or complexity.
>
> W3.3 **(Potential Bias)** The tasks in GridAgent are purposefully designed to effectively evaluate the capabilities of MLLMs, rather than simply solving specific tasks. While the difficulty levels may exhibit a certain artificial nature, they are intentionally crafted to assess the model's performance within well-defined contexts.
>
> W4 **(Human Test)** We addressed the details of the human baseline in **C2 of the Global Response**. Please refer to that section for a comprehensive explanation.

---

> ### Author Response · Authors · 2024-11-23
> **Response to Reviewer SeHM 2**
>
> Q1.1 **(Performance Metrics)** We would like to clarify that Perception Reasoning is a capability defined in our benchmark, not a specific task. For each task, we calculate its success rate and normalize it relative to the human baseline to produce a score between 0 and 1. To evaluate a capability, we compute the weighted average of all task scores associated with that capability to derive a final result. You can see the detailed calculation procedure in Section F of the appendix.
>
> Q1.2 **(Align with Human)** Our evaluation framework compares AI model performance on specific tasks with human baseline performance. By normalizing the model's success rates relative to human performance, we provide a metric to quantify how closely the model’s capabilities align with human cognition benchmarks.
>
> For Perception Reasoning, we assess model performance across all tasks related to this capability, including:
>
> FI/PU: Tasks involving semantic information (FI) or abstract images (PU).
>
> FI/PL/CO*: Tasks that integrate Perception Reasoning with additional elements like memory or planning.
>
> By taking the weighted average of success rates across these tasks, we evaluate the model's perception reasoning ability in a comprehensive and integrated manner. This approach ensures alignment with human cognitive benchmarks while providing a holistic assessment of the model's reasoning capabilities in both single-ability tasks and composite-ability tasks.
>
> Q2 **(Section 5 Confusing)** In the revised version of the paper, we have reorganized Section 5 to align with the sequence of core capabilities outlined in Section 3: Execution, Perception Reasoning, Memory, Learning, and Planning. This restructuring ensures a logical progression, starting with tasks that assess individual capabilities (e.g., Classification for Execution) and advancing to tasks that evaluate combinations of abilities (e.g., Maze* for Memory and Planning). By following this structure, we aim to provide a clearer flow from fundamental tasks to those testing composite capabilities, enhancing the overall coherence of the Methodology and Experimental Analysis sections.
>
> Q3 **(Alternative Comparison)** We discussed about human baseline in **C2** of Global Response. As our benchmark is specifically designed to evaluate MLLMs against foundational cognitive abilities, mirroring stages of human cognitive development.  By comparing MLLMs directly to human performance, we can more effectively gauge their adaptability and developmental potential.  This human-centric comparison is intentional, as it provides a meaningful context to understand the strengths and limitations of MLLMs.

---

> > ### Comment · Reviewer_SeHM · 2024-11-25
> >
> > Thanks for the clarification. I will keep my positive score.

---

> > > ### Author Response · Authors · 2024-11-27
> > > **Response to Reviewer SeHM 3**
> > >
> > > Thank you for your continued support and for keeping your positive score. We appreciate your thoughtful review and are glad that the clarifications addressed your concerns. If you have any further questions or suggestions, please don’t hesitate to reach out.

---

### Official Review · Reviewer_gbMp · 2024-11-08

**Soundness:** 3
**Presentation:** 3
**Contribution:** 3
**Rating:** 6
**Confidence:** 4

**Summary:**

The paper presents GridAgent, a benchmark inspired by Wechsler Intelligence Test to  evaluate five MLLM capabilities: execution, perception reasoning, memory, learning, and planning. To do so, the paper proposes 12 games in 2D grid based environment which are diverse semantic environments, with randomized layouts and varying difficulty to test generalization and robustness of MLLM models. Proposed tasks evaluate multimodal reasoning and understanding abilities of existing MLLMs. Finally, the paper presents quantitative comparison of state-of-the-art proprietary and open-source MLLMs on the GridAgent benchmark and find that on majority of these tasks current MLLMs achieve performance closer to random chance performance. Additionally, the paper presents some preliminary analysis that suggests current MLLMs lack training for image-only perception tasks and it is difficult for these models to solve tasks that require contradictory knowledge from what is learned by these models.

**Strengths:**

1. The paper is well written and easy to follow
2. The tasks in the benchmark are thought through and carefully designed to test the five key capabilities the paper is focused on. I also like how authors created a set of tasks that require composing these capabilities to solve some of the tasks.
3. The experiments section covers most popular state-of-the-art MLLMs apart from Gemini. The zero-shot evaluation results on the tasks clearly demonstrate that these models are lacking zero-shot generalization ability to solve the proposed tasks.

**Weaknesses:**

1. For completeness I’d recommend authors to add results for latest Gemini model in comparison as well.
2. The results demonstrated in the paper are under zero-shot setting. I’d be curious to see how well all of these models ( or a subset of these  for which it is possible given constraints on context length) perform on these tasks with 1-2 in-context examples. We have seen remarkable results with In-Context learning for these long-context models and it’d be good to validate if these tasks can be solved with in-context examples. I believe it should help for SO, FI, and PL tasks. Adding these will also make paper stronger and help highlight the difficulty of tasks in the benchmark
3. Some details about the resolution of image input and how it is fed to each of these models is missing in the paper and supplementary. My understanding is - for all tasks where agent needed to take multiple actions to complete the task the input was fed to the model step by step. Is that the case? If yes, I’d recommend authors to add more details about how the evaluation was done, the resolution of image observations used, etc to the paper.
4. Did the authors ever run experiments with higher resolution image inputs for some of the tasks like Filing? I’d imagine these models should perform better on the task if input image is much higher resolution. If not, I think it would be a good experiment to add to the paper. This would clearly highlight the lack of perception reasoning ability if the input resolution is not a bottleneck.

**Questions:**

I’d also recommend authors to add evaluation examples where some of these models succeeded in supplementary. Examples from the best model should be good to add

I believe the benchmark is interesting and valuable to the community if authors address my concerns I'd be happy to increase my rating.

---

> ### Author Response · Authors · 2024-11-23
> **Response to Reviewer gbMp 1**
>
> **Thank you for acknowledging the clarity and quality of our work.** We appreciate your thoughtful feedback and have provided detailed responses to your questions and concerns below:
>
> W1 **(Add Gemini)** Thank you for the suggestion. We have incorporated the Gemini-1.5-Flash model into our benchmark evaluations for completeness. The results are now included in Section 6 of the revised paper.
>
> W2 **(In Context Learning)** We addressed the details of in-context learning in **C3 of the Global Response**. Please refer to that section for a comprehensive explanation.
>
> W3.1 **(Input Details)** Your understanding is correct: for tasks requiring multiple steps to complete, the input prompt and images were provided to the MLLMs step by step, with the model receiving the concatenated history as context at each step. To clarify this further, we have included a detailed explanation of the evaluation procedure in Section C of the appendix. This section outlines the code structure, prompt design, action choices, and answer decoding process.
>
> W3.2 **(Resolution Information)** Thank you for pointing out the omission of resolution details. We have now added this information in Section B.1 of the appendix. Briefly, the game environment consists of a 9×9 grid, with each grid cell rendered at a resolution of 64×64 pixels (resulting in a total image resolution of 576×576 pixels). All the tested MLLMs were capable of processing images at this resolution without issues.
>
> W4 **(Resolution Experiment)** To address this concern, we conducted additional experiments on the FI task, which relies on strong perception reasoning capabilities, using five different input resolutions ranging from 32×32 to 96×96 per grid. The results indicate that increasing input resolution has a limited impact on model performance for the FI task, with a correlation coefficient of 0.268, suggesting minimal relevance between resolution and performance.
>
> Some MLLMs showed slight improvements at higher resolutions, while others performed worse. This behavior may stem from the nature of the task, which primarily requires an understanding of global features rather than fine-grained pixel-level details. The images used in the task depict simple animals, and the provided options differ significantly in terms of color, spatial information, and other distinguishing features. Thus, the task does not demand models to focus on minute pixel-level details. Once the resolution exceeds a certain threshold (as long as it is not exceedingly small), further increases have negligible impact on performance.
>
> These findings suggest that the performance bottleneck for such tasks lies more in the reasoning capabilities or architectural design of the models than in the input resolution. Different MLLMs may also have their own optimal resolution ranges for tasks of this nature. Detailed results and analysis are included in Section E of the appendix.
>
> |          | 32*32    | 48*48 | 64*64    | 80*80    | 96*96    |
> | -------- | -------- | ----- | -------- | -------- | -------- |
> | GPT-4o   | 0.51     | 0.46  | 0.49     | **0.62** | 0.48     |
> | Gemini   | 0.38     | 0.42  | 0.42     | 0.41     | **0.44** |
> | Qwen2    | 0.29     | 0.40  | **0.47** | 0.43     | 0.45     |
> | Internvl | 0.34     | 0.31  | 0.36     | **0.42** | 0.34     |
> | DeepSeek | **0.45** | 0.43  | 0.39     | 0.39     | **0.45** |
> | Phi3.5   | 0.32     | 0.31  | 0.36     | **0.43** | 0.32     |
> | Llava    | 0.33     | 0.37  | 0.32     | **0.44** | 0.38     |
> | InternLM | 0.32     | 0.34  | **0.42** | 0.37     | 0.35     |
>
> Q1 **(Success Example)** Thanks for your suggestion! We have included successful examples from the best-performing model for each task in Section G of the appendix.

---

> > ### Comment · Reviewer_gbMp · 2024-11-26
> > **Response to authors**
> >
> > Thank you for addressing my concerns and including additional experiments, and missing details!
> >
> > I am inclined towards accepting this paper and I have updated the score to indicate the same

---

> > > ### Author Response · Authors · 2024-11-27
> > > **Response to Reviewer gbMp 2**
> > >
> > > Thank you very much for your kind words and for taking the time to review the updated version of our paper. We are grateful for your thoughtful feedback and are pleased to hear that the additional experiments and clarifications have addressed your concerns.
> > >
> > > We appreciate your updated score and are encouraged by your positive assessment of our work. If you have any further suggestions or questions, please feel free to share.

---

### Official Review · Reviewer_bi4B · 2024-11-09

**Soundness:** 2
**Presentation:** 3
**Contribution:** 2
**Rating:** 5
**Confidence:** 4

**Summary:**

The authors introduce GridAgent, a new benchmark for evaluating "execution", "perception reasoning", "memory", "learning" and "planning" abilities of Multimodal Large Language Models. The authors base this taxonomy on the Wechsler Intelligence Test. The benchmark includes 15 different tasks setups targeting one or more of the abilities.

**Strengths:**

S1: The authors propose a taxonomy for evaluating visual abilities of MLLMs and ground it in existing intelligence test (Wechsler Intelligence Test) setup which has been used to evaluate human cognition

S2: The task setups and scenarios are reproducible and demonstrate the shortcoming of current MLLMs on visual reasoning

**Weaknesses:**

W1: This work provides limited insight compared to previous works. [1] Provides a more in-depth analysis of memory capabilities of MLLMs. Embodied and Computer control benchmarks like [2], [3], [4] provide more real-world insight into the planning and execution abilities of MLLMs in visual setups. [5] provides a detailed analysis of perception reasoning abilities of MLLMs including analysis of Visual, Text and Joint reasoning abilities and failure modes.

W2: Reasoning strategies like Chain-of-Thought [6], Self-consistency [7] are necessary for invoking reasoning abilities of LLMs. Similarly multi-turn tasks (like the ones introduced in this environment) are more suited to agentic formulation of LLMs like ReAct [8], and Reflexion [9]. The results sections lacks these details (did you use COT?) and evaluations of agentic frameworks.

W3: Human evaluation settings are unclear and not well-motivated. Firstly, “Five players (including two authors) participated in the games through the GridAgent interface, with each completing five rounds across all tasks. " It is not well motivated to have authors who are intimately familiar with the task setups to serve as testers. Secondly, "The results show that any adult, once familiar with the rules and putting in serious effort, is fully capable of completing all the games." Where are these results? Lastly, "Consequently, we set the human baseline to 1 for all tasks.” This implies that the human baselines were set to 1.0 without actually completing the same set of tasks that the MLLMs solved. These facts need to be clarified and more details about the human evaluation need to be mentioned.

W4: Lack of Variability studies: Does the performance of MLLMs across multiple trials remain consistent? Authors could evaluate this on a subset of tasks.

W5: Lack of studies about scaling with model sizes: Does the performance of MLLMs on these tasks improve with more model parameters? All considered models are 7B parameters or less providing no information about whether larger model sizes correlated with better performance.

[1] Wang, H., Shi, H., Tan, S., Qin, W., Wang, W., Zhang, T., Nambi, A., Ganu, T., & Wang, H. (2024). Multimodal Needle in a Haystack: Benchmarking Long-Context Capability of Multimodal Large Language Models. arXiv preprint arXiv:2406.11230. https://arxiv.org/abs/2406.11230

[2] Gan, C., Zhou, S., Schwartz, J., Alter, S., Bhandwaldar, A., Gutfreund, D., Yamins, D. L. K., DiCarlo, J. J., McDermott, J., Torralba, A., & Tenenbaum, J. B. (2021). The ThreeDWorld Transport Challenge: A Visually Guided Task-and-Motion Planning Benchmark for Physically Realistic Embodied AI. arXiv preprint arXiv:2103.14025. https://arxiv.org/abs/2103.14025

[3] Xie, T., Zhang, D., Chen, J., Li, X., Zhao, S., Cao, R., Hua, T. J., Cheng, Z., Shin, D., Lei, F., Liu, Y., Xu, Y., Zhou, S., Savarese, S., Xiong, C., Zhong, V., & Yu, T. (2024). OSWorld: Benchmarking Multimodal Agents for Open-Ended Tasks in Real Computer Environments. arXiv preprint arXiv:2404.07972.

[4] Koh, J. Y., Lo, R., Jang, L., Duvvur, V., Lim, M. C., Huang, P. Y., Neubig, G., Zhou, S., Salakhutdinov, R., & Fried, D. (2024). VisualWebArena: Evaluating Multimodal Agents on Realistic Visual Web Tasks. arXiv preprint arXiv:2401.13649.

[5] Guan, T., Liu, F., Wu, X., Xian, R., Li, Z., Liu, X., Wang, X., Chen, L., Huang, F., Yacoob, Y., Manocha, D., & Zhou, T. (2024). HallusionBench: An Advanced Diagnostic Suite for Entangled Language Hallucination and Visual Illusion in Large Vision-Language Models. arXiv preprint arXiv:2310.14566. https://arxiv.org/abs/2310.14566

[6] Wei, J., Wang, X., Schuurmans, D., Bosma, M., Ichter, B., Xia, F., Chi, E., Le, Q., & Zhou, D. (2023). Chain-of-Thought Prompting Elicits Reasoning in Large Language Models. arXiv preprint arXiv:2201.11903. https://arxiv.org/abs/2201.11903

[7] Wang, X., Wei, J., Schuurmans, D., Le, Q., Chi, E., Narang, S., Chowdhery, A., & Zhou, D. (2023). Self-Consistency Improves Chain of Thought Reasoning in Language Models. arXiv preprint arXiv:2203.11171. https://arxiv.org/abs/2203.11171

[8] Yao, S., Zhao, J., Yu, D., Du, N., Shafran, I., Narasimhan, K., & Cao, Y. (2023). ReAct: Synergizing Reasoning and Acting in Language Models. arXiv preprint arXiv:2210.03629. https://arxiv.org/abs/2210.03629

[9] Shinn, N., Cassano, F., Berman, E., Gopinath, A., Narasimhan, K., & Yao, S. (2023). Reflexion: Language Agents with Verbal Reinforcement Learning. arXiv preprint arXiv:2303.11366. https://arxiv.org/abs/2303.11366

**Questions:**

Q1: GPT-4o performs worse than random and other, much smaller MLLMs on PU, PL, and SO tasks. Can you provide a discussion on why this is the case? Can you also check if other models which are larger than 7B also similarly perform worse on these tasks?

Q2: How does the Level of difficulty affect LLMs performance? Could you add results across different difficult levels for those MLLMs?

Q3: Do agentic workflows like ReAct and Reflexion improve the performance of MLLMs in these multi-turn tasks?

Q4: Could you provide more details about how those specific abilities are connected to the Wechsler Intelligence Test?

---

> ### Author Response · Authors · 2024-11-23
> **Response to Reviewer bi4B 1**
>
> Thank you for your detailed and insightful feedback. Below, we provide our responses to your questions and concerns:
>
> W1 **(Limited Insight to Previous Work)** Our work aims to comprehensively evaluate whether current MLLMs have acquired the **fundamental capabilities necessary to perform general tasks in a manner comparable to humans**. In contrast, the benchmarks mentioned in the review primarily focus on **isolated capabilities** or **specialized fields**.
>
> For example, [1] evaluates memory by creating static, context-length-driven inputs through concatenated images, which is only one of the five core capabilities evaluated in GridAgent. Similarly, [2] targets Reinforcement Learning (RL) and Hierarchical Planning Agents, with limited relevance to MLLMs.
>
> Benchmarks such as [3] and [4] focus on system-level agents interacting with web browsers and operating systems, addressing specific application domains rather than general capabilities. Meanwhile, [5] investigates perception reasoning through image-based question-answering tasks, diagnosing phenomena such as language hallucination and visual illusions, but does not evaluate the broader set of reasoning and functional skills we address in GridAgent.
>
> In summary, **GridAgent provides deeper insight by addressing a critical gap: understanding how well current MLLMs generalize across multiple core capabilities in scenarios that mimic the complexity and variability of real-world environments**. This goes beyond diagnosing individual skills or task-specific performance, instead highlighting the overall strengths and weaknesses of MLLMs as cohesive systems.
>
> Therefore, we respectfully disagree with the claim that our work provides limited insight compared to previous benchmarks. On the contrary, our framework bridges an important gap in the literature by enabling holistic, structured evaluation of MLLMs’ general capabilities. We will include a discussion of these complementary works in the revised version of our paper to clarify these distinctions and contributions.
>
> W2 & Q3 **(Agentic Framework)** GridAgent is designed to provide insights into the diverse capabilities of intelligent MLLM models, with our experiments specifically targeting zero-shot reasoning to evaluate the inherent abilities of MLLMs. This approach intentionally avoids relying on heavily engineered prompting strategies or specific agent architectures, such as Chain-of-Thought (CoT), Self-Consistency, ReAct, or Reflexion. By focusing on the raw reasoning and decision-making abilities of MLLMs, we aim to establish a benchmark that offers a fair and consistent evaluation across different models.
>
> Given the rapid pace of advancements in agent frameworks, results derived from specific strategies can quickly become outdated. In contrast, our benchmark is framework-agnostic, designed to remain relevant over time and independent of any particular agentic formulation. This ensures that GridAgent serves as a robust, general-purpose tool for assessing MLLMs’ core capabilities.
>
> We acknowledge the potential value of advanced prompt strategies and agent frameworks, particularly for managing multi-turn tasks. Similar to what **C3** might achieve in Global Response, we anticipate that developers will leverage such approaches to achieve improved results on our benchmark. However, our current focus is on establishing a baseline evaluation of MLLMs’ intrinsic abilities, **providing a foundation upon which such techniques can be further explored.**
>
> W3 **(Human Evaluation)** We addressed the details of the human baseline in **C2 of the Global Response**. Please refer to that section for a comprehensive explanation.
>
> W4 **(Lack of Variability)** To ensure the representativeness and reliability of our experimental results, we set the temperature parameter of each MLLM to 0. This approach guarantees deterministic outputs, compelling the models to select the response with the highest probability and eliminating stochastic variations in the response generation process. Consequently, the MLLMs consistently perform the same operations when presented with the same tasks.
>
> Additionally, each model was evaluated over 500 iterations for every task. These iterations incorporated randomized layouts and objectives, ensuring that the tasks rigorously tested both the generalization and adaptability of the models. This extensive sample size provides robust statistical evidence to support our findings, effectively mitigating potential errors due to chance.
>
> We believe that this comprehensive and carefully designed evaluation process addresses concerns about variability and ensures the reliability of our reported results.

---

> > ### Author Response · Authors · 2024-11-23
> > **Response to Reviewer bi4B 2**
> >
> > W5.1 **(Improve with Model Parameters)** To address this concern, we conducted additional experiments comparing the performance of Qwen2-VL-7B and Qwen2-VL-72B on all Level1 tasks. The results, as added in the revised paper, demonstrate that larger models generally perform better across most tasks. For instance, Qwen2-VL-72B achieves significantly higher success rates on multi-turn tasks that require planning capability such as MA(0.98>0.071), MA*(0.31>0.10) and CO(0.59>0.50). However, we also observed tasks where Qwen2-VL-7B outperformed their larger counterparts. Specifically, in the SE task, the 7B model(0.410) exhibited better performance than the 72B model(0.24). We hypothesize that this may result from the increased complexity of reasoning processes in larger models, which occasionally introduces noise when solving relatively straightforward tasks.
> >
> > |      | CL       | SE       | DE       | MA       | FI       | PU       | MA*      | DE*      | SO       | FI*      | PL       | CO       |
> > | ---- | -------- | -------- | -------- | -------- | -------- | -------- | -------- | -------- | -------- | -------- | -------- | -------- |
> > | 72b  | **0.79** | 0.24     | 0.27     | **0.98** | **0.52** | 0.21     | **0.31** | **0.38** | 0.71     | **0.40** | 0.06     | **0.59** |
> > | 7b   | 0.70     | **0.41** | **0.34** | 0.71     | 0.47     | **0.25** | 0.10     | **0.38** | **0.72** | 0.35     | **0.14** | 0.50     |
> >
> > W5.2 **(Larger Model)** Furthermore, we incorporated additional experiments with the recently released Gemini-1.5-Flash model. Consistent with our findings, larger models such as GPT-4o, Gemini-1.5-Flash, and Qwen2-VL-72B generally outperform smaller models (≤7B) on a majority of tasks. However, differences persist across task types, reaffirming that performance can be influenced by both model size and task characteristics. In the PL and SO tasks, the results are particularly striking, with the larger models showing significantly lower accuracy than the smaller models. A possible explanation of this phenomenon can be found in **Q1**, and more raw data statistics for all models can be seen in Section B.4 of the appendix.
> >
> > Q1.1 **(GPT worse)** During our review of the PU/PL/SO tasks, we identified an issue in the initial implementation where some Roman numerals in the options were not correctly parsed, which partly affected the results. This has been addressed in the updated version. In the new PU results, GPT-4o performed similarly to other large models, around baseline. For the PL and SO tasks (two tasks with lower human baselines), GPT-4o is still underperforming other models.
> >
> > |          | PU       | PL       | SO       |
> > | -------- | -------- | -------- | -------- |
> > | GPT-4o   | **0.26** | **0.06** | **0.58** |
> > | Gemini   | 0.24     | 0.16     | 0.47     |
> > | Qwen2    | 0.25     | 0.14     | 0.72     |
> > | InternVL | 0.22     | 0.11     | 0.96     |
> > | DeepSeek | 0.27     | 0.19     | 0.57     |
> > | Phi3.5   | 0.25     | 0.12     | 0.55     |
> > | Llava    | 0.25     | 0.12     | 0.42     |
> > | InternLM | 0.25     | 0.17     | 0.44     |
> >
> > Q1.2 **(Why GPT Worse)** Both PL and SO tasks evaluate the MLLM's learning capability, and the rules to be learned in both tasks are counterintuitive. For instance, in the SO task, agents must rank animals according to a rule such as, "In this world, the faster the animal, the lighter it is," and then sort animals by weight. We added a Level 1 version of SO, requiring the ranking of only two animals, to test whether the agents understood basic comparative relationships in the new set of rules. However, GPT-4o's performance remained suboptimal on level 1 test. We found that GPT-4o had difficulty accepting that a horse would be lighter than a pig, as it conflicted with its real-world knowledge that horses are heavier than pigs. This conflict caused significant confusion in correctly ranking the animals according to the new rule
> >
> > Q1.3 **(Other Large Models)** At the same time, the newly tested model also showed lower accuracy on the PL task, demonstrated challenges in large models adapting to new tasks that require rule learning that conflicts with prior knowledge.
> >
> > Q2 **(Difficulty Level)** We addressed the details of the difficulty levels in **C1 of the Global Response**. Please refer to that section for a comprehensive explanation.

---

> > > ### Author Response · Authors · 2024-11-23
> > > **Response to Reviewer bi4B 3**
> > >
> > > Q4 **(Wechsler Intelligence Test)** Our MLLM capability design is informed by the Wechsler Intelligence Scale, particularly the Wechsler Intelligence Scale for Children (WISC)[6]. MLLMs are still in the early stages of development. While assessments for adults typically emphasize specialized knowledge or skills, children's tests focus more on fundamental cognitive abilities that underpin general intelligence. By evaluating MLLMs against these foundational abilities, we can more effectively gauge their adaptability and developmental potential, mirroring the stages of human cognitive growth. This approach allows us to gain a comprehensive understanding of MLLMs' strengths and limitations, providing clear guidance for improvements and ultimately contributing to the advancement of AGI.
> > >
> > > Dr. David Wechsler conceptualized intelligence as a composite of distinct cognitive abilities. In the WISC framework, these abilities are organized into five primary domains: the Verbal Comprehension Index (VCI), the Visual Spatial Index (VSI), the Fluid Reasoning Index (FRI), the Working Memory Index (WMI), and the Processing Speed Index (PSI). We used these domains as a reference to define MLLM's task performance abilities, categorizing them into five areas: Execution, Perception Reasoning, Memory, Learning, and Planning.
> > >
> > > Here's how each of these MLLM capabilities connects to the specific cognitive domains measured by the Wechsler Intelligence Test:
> > >
> > > Execution: Connected to the Processing Speed Index (PSI). In the WISC, PSI assesses how quickly and accurately a child can complete simple tasks. For example, a child might be asked to match as many pairs of symbols and digits as possible in a given time. However, for MLLMs, processing speed is less of a concern because we are mainly examining reasoning tasks, focusing on accuracy rather than efficiency. Instead, we focus on evaluating the accuracy and correctness of their actions in execution tasks. These tasks assess whether the model can correctly interpret instructions, make decisions, and act accordingly, independent of the time taken.
> > >
> > > Perception Reasoning: Aligns with the Visual Spatial Index (VSI) and the Fluid Reasoning Index (FRI). VSI in the WISC measures a child's ability to interpret and organize visual information, while FRI assesses problem-solving and abstract thinking abilities. In MLLMs, Perception Reasoning involves interpreting input data—be it visual, textual, or auditory—and making logical inferences. This capability allows the model to understand context, recognize patterns, and establish relationships between different pieces of information.
> > >
> > > Memory: Corresponds to the Working Memory Index (WMI). WMI evaluates a child's capacity to temporarily hold and manipulate information. Similarly, Memory in MLLMs involves retaining contextual information over interactions and recalling relevant data when needed. This capability ensures coherent and contextually appropriate responses, allowing the model to maintain consistency and relevance in extended dialogues or tasks.
> > >
> > > Learning: Underpinned by the Verbal Comprehension Index (VCI). VCI measures a child's ability to understand and process verbal information, which is essential for acquiring new knowledge. In MLLMs, Learning refers to the ability to absorb new information, adapt to new data, and improve performance over time. By ensuring strong verbal comprehension, MLLMs can more effectively learn from interactions and adapt their responses accordingly.
> > >
> > > Planning: Analogous to executive functions, which, while not explicitly isolated in the WISC, are essential for complex cognitive tasks. Planning involves strategizing and sequencing actions to achieve specific goals, considering potential outcomes. In MLLMs, this ability enables the model to generate coherent, goal-oriented responses or actions, reflecting the kind of higher-order thinking skills that support children's performance across various WISC subtests.
> > >
> > > The design of each capability fully considers the differences between humans and MLLMs, ensuring that our approach leverages established cognitive frameworks while adapting them to the unique functionalities of MLLMs.  This alignment not only provides a robust structure for assessing MLLM performance but also facilitates targeted enhancements in their development.  A more detailed explanation is provided in Section A.2 of the appendix.

---

> > > > ### Author Response · Authors · 2024-11-23
> > > > **Response to Reviewer bi4B 4**
> > > >
> > > > [1] Wang, H., Shi, H., Tan, S., Qin, W., Wang, W., Zhang, T., Nambi, A., Ganu, T., & Wang, H. (2024). Multimodal Needle in a Haystack: Benchmarking Long-Context Capability of Multimodal Large Language Models. arXiv preprint arXiv:2406.11230. https://arxiv.org/abs/2406.11230
> > > >
> > > > [2] Gan, C., Zhou, S., Schwartz, J., Alter, S., Bhandwaldar, A., Gutfreund, D., Yamins, D. L. K., DiCarlo, J. J., McDermott, J., Torralba, A., & Tenenbaum, J. B. (2021). The ThreeDWorld Transport Challenge: A Visually Guided Task-and-Motion Planning Benchmark for Physically Realistic Embodied AI. arXiv preprint arXiv:2103.14025. https://arxiv.org/abs/2103.14025
> > > >
> > > > [3] Xie, T., Zhang, D., Chen, J., Li, X., Zhao, S., Cao, R., Hua, T. J., Cheng, Z., Shin, D., Lei, F., Liu, Y., Xu, Y., Zhou, S., Savarese, S., Xiong, C., Zhong, V., & Yu, T. (2024). OSWorld: Benchmarking Multimodal Agents for Open-Ended Tasks in Real Computer Environments. arXiv preprint arXiv:2404.07972. https://arxiv.org/abs/2404.07972
> > > >
> > > > [4] Koh, J. Y., Lo, R., Jang, L., Duvvur, V., Lim, M. C., Huang, P. Y., Neubig, G., Zhou, S., Salakhutdinov, R., & Fried, D. (2024). VisualWebArena: Evaluating Multimodal Agents on Realistic Visual Web Tasks. arXiv preprint arXiv:2401.13649. https://arxiv.org/abs/2401.13649
> > > >
> > > > [5] Guan, T., Liu, F., Wu, X., Xian, R., Li, Z., Liu, X., Wang, X., Chen, L., Huang, F., Yacoob, Y., Manocha, D., & Zhou, T. (2024). HallusionBench: An Advanced Diagnostic Suite for Entangled Language Hallucination and Visual Illusion in Large Vision-Language Models. arXiv preprint arXiv:2310.14566. https://arxiv.org/abs/2310.14566
> > > >
> > > > [6] Park, S.E., Demakis, G.J. (2017). Wechsler Intelligence Scale for Children. In: Zeigler-Hill, V., Shackelford, T. (eds) Encyclopedia of Personality and Individual Differences. Springer, Cham. https://doi.org/10.1007/978-3-319-28099-8_1035-1

---

> > > ### Comment · Reviewer_bi4B · 2024-11-27
> > > **Thanks for the clarifications**
> > >
> > > The authors have addressed most of my questions and concerns. I believe the addition of human baseline results, experiments with ICL examples, and comparison of different parameter-sized models warrants an increase in rating. I have updated my score to a 5.

---

> > > > ### Author Response · Authors · 2024-11-28
> > > > **Response to Reviewer bi4B 5**
> > > >
> > > > Thank you for your thoughtful and constructive feedback. We are pleased to hear that the supplementary experiments  have effectively addressed your concerns. We greatly appreciate your updated score and the careful evaluation of our work.
> > > >
> > > > If you have any further suggestions or inquiries, please feel free to contact us.

---

### Author Response · Authors · 2024-11-23
**Global Response 1**

We sincerely thank all reviewers for their thoughtful feedback. We are encouraged that our work is recognized as valuable (gbMp) and significant (SeHM), our tasks are considered thoughtful (gbMp), and our scenarios are deemed reproducible (bi4B). In this rebuttal, we provide a point-by-point response to all comments and include additional experiments conducted on the GridAgent benchmark.

Below, we summarize the major additions and adjustments made to the paper:

In the main paper:

- We incorporated the Gemini-1.5-Flash model into our benchmark evaluations to enhance completeness.
- We rephrased Section 5 (Tasks in GridAgent) to align with the sequence of capabilities introduced in Section 3. The revised section begins with tasks evaluating individual capabilities and progresses to tasks assessing composite capabilities, ensuring a clear hierarchy and logical flow.
- We updated Table 3 in Section 6 to replace the original success rate data with scores normalized to the human baseline. This adjustment enables readers to better compare the performance of different models with that of humans.

In the appendix:

- We introduced a new Section A to describe the connection between our benchmark and the Wechsler Intelligence Test, explaining how we designed the five core capabilities (Execution, Perception Reasoning, Memory, Learning, Planning) to evaluate MLLMs.
- A new Section B was added to provide additional experimental details, including the resolution setup and parameter selection during the inference stage of MLLMs.
- Section C was reorganized to specify the evaluation procedure of our benchmark, detailing the code structure, prompt design, action choices, and the decoding process for extracting answers from MLLMs.
- In Section E, we added new experiments to further support our work, including analyses of input resolution, model size, and the impact of in-context learning.
- We updated Section G (Task Information) by replacing the original human examples with successful examples from the best-performing model for each task.

These additions and adjustments strengthen the clarity, rigor, and reproducibility of our work and address the insightful feedback provided by the reviewers.

Furthermore, we would like to highlight some important changes and additional experiments:



C1 **(Difficulty Level)** Initially, we observed that most MLLMs performed reliably only at Level 1, with performance dropping significantly, often approaching the random baseline at Levels 2 and 3. Based on these findings, we focused on Level 1 results in the main paper to highlight scenarios where the models demonstrated meaningful capabilities. Additionally, some tasks originally did not differentiate between difficulty levels. In our latest version, every task includes three difficulty levels: Level 1 represents the simplest version of the task, while Levels 2 and 3 introduce progressively greater complexity through additional challenges. For tasks where MLLMs generally performed poorly, we also adjusted the difficulty settings to better align with the models’ current capabilities, ensuring a more balanced evaluation. Specific changes are detailed below, with additional information on difficulty levels provided in Section G.

- PU: Considering that the original PU was difficult (need to complete 4 puzzles), the new difficulty level 1/2/3 was adjusted to complete 1/2/3 puzzle(s) separately.
- SO:  Considering that the original SO was difficult (need to sort 3 animals according to the rule), the new difficulty level 1/2/3 was adjusted to rank 2/3/4 kinds of animals.
- PL: Considering that the original PL had only one difficulty (need to place an item according to the opposite instruction), the new difficulty level was adjusted to  perform operations according to simple/medium/hard rule.
- FI & FI*: Considering that the original FI and FI\* had only one difficulty (need to complete 1 missing piece of an animal image), the new difficulty level 1/2/3 was adjusted to complete 1/2/3 missing piece(s) separately. Besides, we removed some of confusing options with only marginal information.
- MA & MA*: Considering that the original MA and MA\* was difficult (always appear 3 pairs of doors and keys), the new difficulty level 1/2/3 was adjusted to appear 1/2/3 pairs of doors and keys in the scene.

We recognize the importance of presenting the complete range of results to illustrate the impact of increasing task difficulty. Accordingly, we have now included the performance results across all difficulty levels (levels 1, 2, and 3) in Section B.4 of the appendix.

---

> ### Author Response · Authors · 2024-11-23
> **Global Response 2**
>
> C2 **(Human Baseline)** We acknowledge the limitations in our initial human evaluation setup, including the involvement of authors as testers and the default setting of the human baseline to 1.0. To address these concerns, we conducted a new round of human evaluations with 18 university students who had no prior association with the development of the benchmark. The participants were divided into three groups of six, with each group assigned to test one of the three difficulty levels. To minimize the potential influence of repeated attempts and accumulated experience, which could artificially inflate success rates, each participant completed only a single iteration of their assigned tasks. This approach ensured a more objective assessment of human performance.
>
> The results of these evaluations (CL1 means CL task of Level1) confirmed that the tasks are not inherently difficult for humans but highlighted common challenges such as misreading instructions, forgetting prior information, or misunderstanding task rules. These findings reinforce the benchmark's ability to test capabilities that require careful attention and reasoning.
>
> | CL1  | CL2  | CL3  | SE1  | SE2  | SE3  | DE1  | DE2  | DE3  | MA1  | MA2  | MA3  |
> | ---- | ---- | ---- | ---- | ---- | ---- | ---- | ---- | ---- | ---- | ---- | ---- |
> | 1.00 | 1.00 | 0.83 | 1.00 | 1.00 | 0.89 | 1.00 | 1.00 | 1.00 | 1.00 | 1.00 | 1.00 |
>
> | FI1  | FI2  | FI3  | PU1  | PU2  | PU3  | MA*1 | MA*2 | MA*3 | DE*1 | DE*2 | DE*3 |
> | ---- | ---- | ---- | ---- | ---- | ---- | ---- | ---- | ---- | ---- | ---- | ---- |
> | 0.94 | 0.94 | 0.89 | 1.00 | 1.00 | 0.94 | 0.94 | 0.83 | 0.72 | 1.00 | 1.00 | 0.94 |
>
> | SO1  | SO2  | SO3  | FI*1 | FI*2 | FI*3 | PL1  | PL2  | PL3  | CO1  | CO2  | CO3  |
> | ---- | ---- | ---- | ---- | ---- | ---- | ---- | ---- | ---- | ---- | ---- | ---- |
> | 0.83 | 0.72 | 0.67 | 0.89 | 0.89 | 0.83 | 0.72 | 0.67 | 0.44 | 1.00 | 1.00 | 1.00 |
>
> To improve the interpretability of our results, we normalized the success rates of the MLLMs by comparing them to the observed human success rates, rather than relying on absolute values. This adjustment provides a more robust and meaningful evaluation of MLLMs’ performance relative to human capabilities. Detailed descriptions of the updated human evaluation process and results are provided in Section D of the appendix.

---

> ### Author Response · Authors · 2024-11-23
> **Global Response 3**
>
> C3 **(Prompt Design + In Context Learning)** Our benchmark is primarily designed to provide a standardized framework for testing MLLMs, leaving choices such as input formatting (e.g., incorporating in-context examples) to the discretion of developers and researchers. That said, we appreciate the value of exploring in-context examples to assess their impact on model performance and have included such evaluations in the revised version for the SO, PL, and FI tasks. These tasks were selected because they require counterintuitive rule learning (SO, PL) or spatial reasoning to infer missing components (FI).
>
> In our example prompts, we demonstrate an entire process the agent follows to accomplish the task: First, we introduce the arrangement of items in the scene to provide background information; next, we state the goal of the task so the agent can understand what it needs to achieve; finally, we explain the reasoning behind each action the agent takes, showing how it completes the task step by step.
>
> |          | Origin(FI) | With Example(FI) | Origin(SO) | With Example(SO) | Origin(PL) | With Example(PL) |
> | :------: | :--------: | :--------------: | :--------: | :--------------: | :--------: | :--------------: |
> |  GPT-4o  |    0.49    |    **0.59↑**     |    0.58    |    **0.76↑**     |    0.06    |    **0.31↑**     |
> |  Gemini  |    0.42    |    **0.56↑**     |    0.47    |    **0.69↑**     |    0.16    |    **0.17↑**     |
> |  Qwen2   |    0.47    |       0.36       |    0.72    |    **0.91↑**     |    0.14    |       0.11       |
> | InternVL |    0.36    |    **0.41↑**     |    0.96    |       0.63       |    0.11    |    **0.17↑**     |
> | DeepSeek |    0.39    |    **0.41↑**     |    0.57    |    **0.98↑**     |    0.19    |       0.10       |
> |  Phi3.5  |    0.36    |    **0.44↑**     |    0.55    |    **0.91↑**     |    0.12    |       0.10       |
> |  Llava   |    0.32    |    **0.46↑**     |    0.42    |    **0.84↑**     |    0.12    |    **0.17↑**     |
> | InternLM |    0.42    |       0.36       |    0.44    |    **0.72↑**     |    0.17    |       0.11       |
>
> Our results show varying effects:
>
> SO Task: Performance improved significantly for most models, with DeepSeek, for example, increasing from 0.57 to 0.98.
>
> FI Task: Modest improvements were observed; Gemini improved from 0.42 to 0.56.
>
> PL Task: Performance remained unchanged or even deteriorated for some models.
>
> Interestingly, we found that adding examples sometimes led to confusion in certain models:
>
> For the InternVL model in the SO task, performance dropped from 0.96 to 0.63 after adding examples. Analysis revealed that longer prompts caused the model to struggle with extracting the intended question and returning an appropriate action letter. Instead, it provided unrelated answers, indicating difficulty handling extended inputs effectively.
>
> Similarly, the Qwen2 model's performance in the FI task declined. A noticeable increase in selecting "place piece in backpack C at grid I" was observed—this was the correct answer in the provided example. This suggests that the model overly focused on reproducing the example's final result while neglecting the reasoning process, reflecting a limitation in its learning ability when handling longer prompts.
>
> These findings demonstrate that while in-context examples can significantly enhance performance for tasks requiring complex rule understanding or reasoning (e.g., SO), their effectiveness varies across task types and models. They also highlight challenges some models face in adapting to longer prompts or maintaining focus on the task at hand. We believe these observations strengthen the paper by offering additional insights into model behavior and task difficulty within the benchmark.

---

### Meta-Review · Area_Chair_iNHm · 2024-12-21

**Metareview:**

The paper proposes a synthetic 2D grid based multimodal benchmark to evaluate the different abilities like planning, learning new knowledge during interaction, memory, task execution. These abilities seem to be inspired from the  Wechsler Intelligence Test. The authors evaluated multiple closed-source and open-weights model including GPT4o, Gemini, QwenVL2-72b.

The task setups and scenarios are reproducible and demonstrate shortcoming of current MLLMs. Some findings (for example, learning new knowledge which conflicts with the models' prior knowledge is difficult to learn) are insightful and could be beneficial to the community.

However, several concerns raised during the review and during a closer inspection of the paper remain.
- Reviewer bi4B pointed that the paper uses vanilla models without using Chain-of-Thought, ReACT and Reflexion style models. While these strategies are quickly evolving, they have found to be useful in many benchmarks, and will provide insights on how to more effectively use MLLMs. When presenting a new benchmark, it's important to show results on well-established / popular techniques to extract the most out of existing MLLMs. This provides a reasonable baseline for the benchmark, and will encourage the community to innovate beyond current techniques.
- The work provides limited insights compared to previous works. While no work tests all the described capabilities in a single benchmark, there are several existing benchmarks (specially in the RL community) which test similar or related capabilities like perceptual understanding, action grounding and planning (Reivewer bi4B).
- The paper will benefit from improved writing. While the paper writing improved during the rebuttal period, the paper still doesn't concretely show the agent's trajectory (prompt, task, responses) in one single place either in the main manuscript or supplementary

**Additional Comments On Reviewer Discussion:**

During the rebuttal period,

- Authors were requested to conduct a more thorough human studies. The authors conducted another round of human evaluation with a larger pool, who had no prior knowledge of the benchmark. These evaluations provided a more objective assessment of the benchmark, and helped put the model performance in context. The authors found that humans made errors due to misreading instructions or task rules. We encourage the authors to further tune these instructions, to minimize these human errors to provide a more effective human baseline.
- Multiple reviewers requested improved baselines (using in-context learning, ReACT or Reflexion style baselines). Authors performed additional experiments with in-context examples and showed significant improvements for most tasks and models. We encourage the authors to provide these in the next version of the manuscript, but also use stronger baselines that use chain-of-thought, self-consistency, reflexion style techniques which have been proven effective in the past.

---

### Decision · Program_Chairs · 2025-01-22

Reject